

**Surface Observation Constrained High Frequency Coal**
**Mine Methane Emissions in Shanxi China Reveal More**
**Emissions than Inventories, Consistency with Satellite**
**Inversion**
Fan Lu[1], Qin Kai[1*], Jason Blake Cohen[1*], Qin He[1], Pravash Tiwari[1], Wei Hu[1], Chang
Ye[1], Yanan Shan[1], Qing Xu[1], Shuo Wang[1], Qiansi Tu[2]
[1]Jiangsu Key Laboratory of Coal-Based Greenhouse Gas Control and Utilization, School of Environment
and Spatial Informatics, China University of Mining and Technology, Xuzhou, China
[2]School of Mechanical Engineering, Tongji University, Shanghai, China
*Correspondence to*: Kai Qin (qinkai@cumt.edu.cn) ; Jason Blake Cohen (jasonbc@alum.mit.edu)
**Abstract.** This work focuses on Changzhi, Shanxi China, a city and surrounding rural region with one
of the highest atmospheric concentrations of methane ($CH_4$) world-wide (campaign-wide
minimum/mean/standard deviation/max observations: 2.0, 2.9, 1.3, and 16 ppm) due to a rapid increase
in the mining, production, and use of coal over the past decade. An intensive 15-day surface observation
campaign of $CH_4$ is used to drive a new analytical, mass-conserving method to compute and attribute
$CH_4$ emissions. Observations made in concentric circles at 1km, 3km, and 5km around a high production
high gas coal mine yielded emissions of 0.73, 0.28, and 0.15 ppm min$^{-1}$ respectively. Attribution used a
2-box mass conserving model to identify the known mine's emissions from 0.042-5.3 ppm min$^{-1}$, and a
previously unidentified mine's emission from 0.22-7.9 ppm min$^{-1}$. These results demonstrate the
importance of simultaneously quantifying both the spatial and temporal distribution of $CH_4$ to better
control regional-scale $CH_4$ emissions. Results of the attribution are used in tandem with observations of
boundary layer height to quantify policy-relevant emissions from the two coal mines as 13670±7400 kg
h$^{-1}$ and 5070±2270 kg h$^{-1}$ respectively. Both mines display a fat tail distribution, with respective 25th,
median, and 75th percentile values of [870, 7500, 38700] kg h$^{-1}$ and [431, 1590, 7000] kg h$^{-1}$. These
findings are demonstrated to be higher than $CH_4$ emissions from equivalent oil and gas operations in the
USA, with one about double and the other similar to day-to-day emissions inverted over 5-years using
TROPOMI over the same region.



**1. Introduction**

Emissions of Methane (CH₄) contribute the second most to direct anthropogenic longwave radiative forcing (Etminan et al., 2016; Li et al., 2022). Since CH₄ has a lifetime from 9.5 to 12.5 years (Li et al., 2022; Prather et al., 2012), controlling CH₄ emissions can provide an opportunity to mitigate peak loading and slow the rate of net global warming. Fossil fuel CH₄ is one of the largest sources of anthropogenic CH₄ emissions (Kirschke et al., 2013; Saunois et al., 2020a). Since China is the world's largest producer and consumer of coal (Bournazian, 2016), Coal mines CH₄ (CMM) possibly contributes up to 33%-40% of China's total CH₄ emissions (Janssens-Maenhout et al., 2017; Miller et al., 2019; Peng et al., 2016). Although China enacted CMM regulations in 2010, CMM continues grow (Kerr and Yang, 2009; Miller et al., 2019). CH₄ emission estimates are highly uncertain in both space and time (Brandt et al., 2014; Saunois et al., 2020b). They also generally have a fat tail distribution, wherein a small number of samples have extremely large emissions that overwhelm emissions under average conditions (Duren et al., 2019; Plant et al., 2022). For these reasons, new approaches to quantify, reduce uncertainty, and attribute CH₄ emissions are necessary and can provide support for policies aiming to control and mitigate CMM (Cao, 2017).

Bottom-up (BU) quantification of emissions requires a priori knowledge of source locations and diversity, which tends to not represent real-world conditions. Top-down (TD) approaches analyze concentration data with improving accuracy (Allen, 2014; Rigby et al., 2019; Varon et al., 2018; Vaughn et al., 2018), specifically combining surface (Heerah et al., 2021; Katzenstein et al., 2003; Shi et al., 2023), aircraft (Karion et al., 2013; Shi et al., 2022; Tong et al., 2023; Vinković et al., 2022), and/or satellite (Wecht et al., 2014) CH₄ observations with atmospheric models. Some TD approaches use physically realistic but complex chemical transport models (Bloom et al., 2017) , others use plume models (Goldsmith et al., 2012), and others still use data driven approaches (Buchwitz et al., 2017). Uncertainties are rarely assessed holistically or in detail (Cohen and Prinn, 2011; Cohen and Wang, 2014).

Airborne remote sensing is a highly technical and costly approach to record CH₄ fluxes from landfills, coal basins, and oil and gas production (Krautwurst et al., 2021; Krautwurst et al., 2017; Kuhlmann et al., 2023), which suffers from not being able to monitor CH₄ emissions over long periods of time or in regions where the source is not well constrained (Brandt et al., 2014; Gorchov Negron et al., 2020; Hiller et al., 2014; Mehrotra et al., 2017; Molina et al., 2010). Satellite remote sensing can measure CH₄ under specific orbits where the source is known and identified (Jacob et al., 2016; Jacob et



al., 2022; Plant et al., 2022; Varon et al., 2018; Zhang et al., 2020), but only after being calibrated by
upward looking remotely sensed measurements (Tu et al., 2022), and only when the atmosphere is rain,
cloud and aerosol free (Cohen and Prinn, 2011; Reuter et al., 2019; Sadavarte et al., 2021). TROPOMI
and GOSAT have both been shown to be data-rich at times (Butz et al., 2012; Hu et al., 2018; Jacob et
al., 2016), but severely limited at other times (Butz et al., 2012; Kuze et al., 2009). Even when these
satellites have sufficient data to compute emissions from other species, frequently $CH_4$ cannot be
computed (Li et al., 2023; Qin et al., 2023b) due to insufficient signal strength, and uncertainties which
are both non-understood and mis-constrained (Povey and Grainger, 2015).

Ground-based remote sensing provides higher accuracy versus satellite observations (Heerah et al.,

2021; Luther et al., 2022; Tu et al., 2022). EM27/SUN measurements have approximated $CH_4$ emissions
in Poland (Luther et al., 2019; Luther et al., 2022). However, these instruments are expensive, require
calibration, and have limited data collection due to solar signal strength.

This work employs a high-frequency surface-based observation platform of $CH_4$ concentration

which is portable, economical, and unaffected by most environmental factors. The observations are
combined with a new mass-conserving methodology based on temporal transformation of the spatially
derived mass-conserving framework successfully applied to $NO_2$ (Li et al., 2023; Qin et al., 2023b). This
work focuses on Shanxi, one of the densest coal mining regions in the world, accounting for
approximately 10% of total global coal production (Lin and Liu, 2010; Qin et al., 2023a). Continuous
observations were made around known coal mines, unknown sources, and of background conditions.
High-frequency emissions calculated using these data were used to drive a 2-box model to attribute
emissions to the known mine and a second low production mine previously thought insignificant. The
results provide insights into the spatial distribution of $CH_4$ emissions, demonstrate rapid adoption of
practical methods globally, and enable source attribution.
**2. Methods and Data**
**2.1 Study Site and Campaign Design**

Changzhi, Shanxi is located in a basin, with coal mines densely distributed throughout both flat

central regions and around the mountainous edges (Figure 1), many of which are classified as high $CH_4$
emitting mines. Due to this combination, province-wide background $CH_4$ concentrations are very high
and have large variation in time. This study mainly focuses on two coal mines: one mine is classified as



having high amounts of CH$_4$ per unit of production and an annual coal production of 4 million tons (CM-
A), and the other is unclassified for CH$_4$ per unit of production and having an annual coal production of
3 million tons (CM-B) (Qin et al., 2023a). Observations were positioned along concentric circles located
1km, 3km, and 5km from CM-A, over an approximation of the four ordinal directions: east, west, south,
north (Figure 2). All locations were planned to be far away from known anthropogenic sources, leading
to a net total 12 measurement points. As later discovered, CM-B is located approximately 1km southwest
from the measurement point located at 5km west.

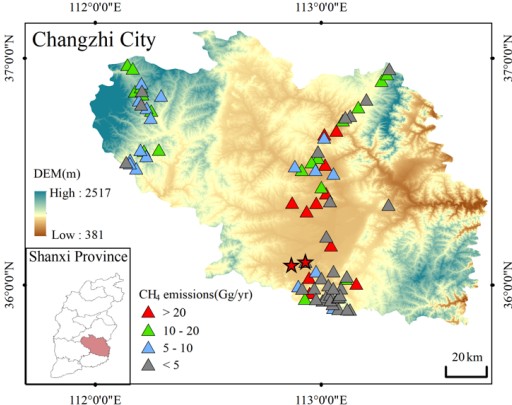


**Figure 1. Topographic map of Changzhi, and its location in Shanxi Province (bottom left). The triangles**
**represent the locations of all individual coal mines (including underground and abandoned mines), where the**
**triangle color represents the emissions amount: high (red), middle (green), low (blue), and very-low (grey).**
**The red stars represent the two coal mines in this work.**

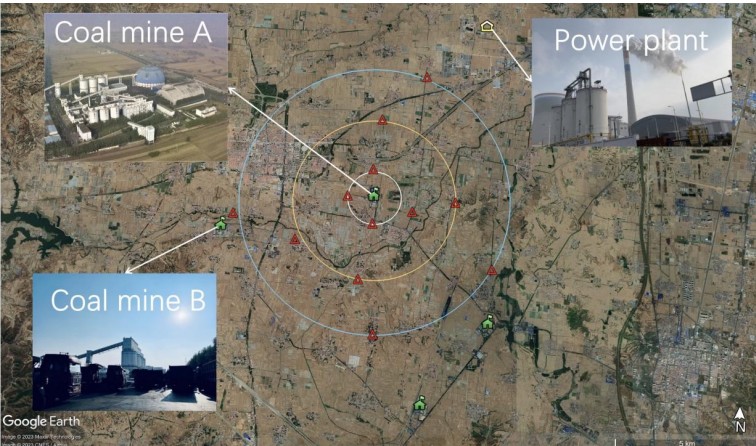


**Figure 2. Locations of four individual coal mines (Green filled houses), a power plant (Yellow outlined house),**
**and the 12 observation locations presented in this work (red double-outlined triangles). Distance from CM-A**



**are given as concentric circles at 1km (white), 3km (yellow), and 5km (blue).**
**2.2 Measuring CH$_4$ concentration**
Atmospheric CH$_4$ concentrations at 5m above the surface were observed daily at 1 Hz from 8:30 to
17:00 local time in August 2022 using two portable greenhouse gas analyzers (LGR-915-0011, California,
USA). Three different locations (1 km, 3 km and 5 km) were selected daily along a single direction from
the CM-A, allowing a more consistent and precise calculation of the spatial gradient (Table 1). In order
to reduce the time errors, using two portable greenhouse gas analyzers randomly selecting the three
observation  points during the daily measurements and without fixed sequence.
The CH$_4$ data was averaged minute-by-minute to match observed wind data, and subsequently used
to compute CH$_4$ emissions. As show in Figure 3, the CH$_4$ concentration data is highly correlated with
rapid changes in both the wind speed and direction.
**Table 1. Detailed Information Summarizing the Dates, Times, and Locations Observed in the Field**

| Date | Direction of measured site | Sample period (min) | | |
|---|---|---|---|---|
| | | 1 km | 3 km | 5 km |
| 10 August 2022 | North | 88 | 35 | 47 |
| 12 August 2022 | North | 78 | 66 | 61 |
| 13 August 2022 | North | 168 | 181 | 178 |
| 14 August 2022 | North | 125 | 128 | 122 |
| 15 August 2022 | North | 121 | 122 | 119 |
| 17 August 2022 | North, West | 122, 124 | 119, 120 | 120, 117 |
| 18 August 2022 | North, West | 143, 119 | 135, 118 | 125, 120 |
| 19 August 2022 | East | 126 | 139 | 140 |
| 21 August 2022 | North | 140 | 127 | 125 |
| 22 August 2022 | East, South | 129, 123 | 124, 133 | 121, 124 |
| 23 August 2022 | South | 126 | 129 | 129 |

Observations made in clean locations with a wind direction not from the mine are subsequently
considered for background sites. The lowest and least variable CH$_4$ observations are found on August 23
in the south (2.08±0.08 ppm) (Figure 3). It is important to note that although this site has the minimum
concentrations observed in this work, these values are significantly higher than the global latitude-band
background. Three other locations and days were observed with relatively low mean and not significantly
large variation: August 19 in the east (2.63±0.35 ppm), August 22 in the east (2.65±0.51 ppm), and
August 22 in the south (2.60±0.55 ppm) (Figure 3). These results imply that the practice adopted by the
community to separate a plume from the global latitude band or climatological background state is not
applicable in the locations sampled in this paper (Buchwitz et al., 2017; Irakulis-Loitxate et al., 2021;
Lauvaux et al., 2022; Sadavarte et al., 2021). For this reason, a new quantitative approach is presented
to understand and quantify what is actually a source and what is not. This approach is applicable under
conditions both encountered globally as well as those under the uniquely high and variable conditions
observed herein.

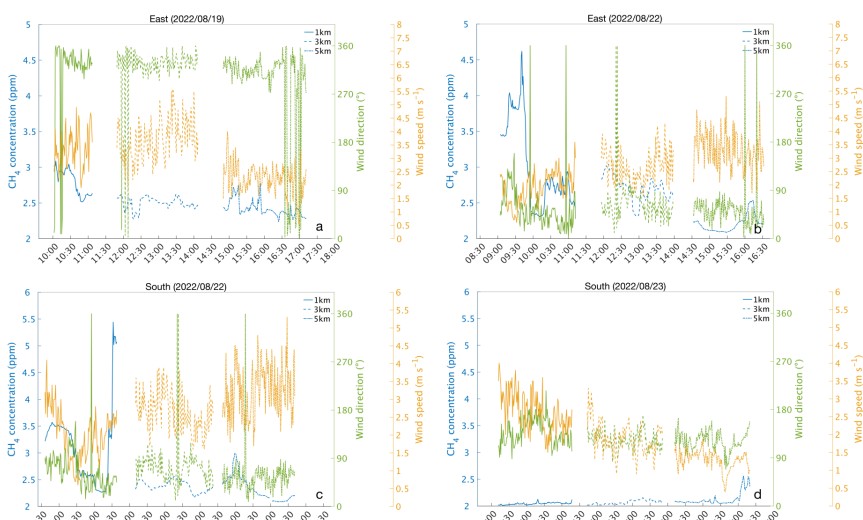

**Figure 3. Time series of CH$_4$ concentration (ppm) (blue), wind speed (m s$^{-1}$) (yellow) and wind direction [º]**

**(orange lines) measured at 1km (solid), 3km (dashed) and 5km (dash-dot) located east (top) and south (bottom)**

**of CM-A on three different days.**

**2.3 Meteorological Data**

The wind speed and direction were obtained from local meteorological stations with a temporal
frequency of 1min. As show in Figures 4 and 5 the overall wind was dominated by a southerly direction
(38.0% of observations between 150° and 210°) and found to be moderately slow (69.9% of observations
were between 1 m s$^{-1}$ and 4 m s$^{-1}$). The 10$^{th}$ and 90$^{th}$ percentiles of wind direction (54° and 312°) and
wind speed (1 m s$^{-1}$ and 5.1 m s$^{-1}$) respectively, indicate that high frequency sampling reveals a small
number of relatively large changes are observed, which are expected to lead to a "fat-tail" type of
distribution of subsequently computed CH$_4$ emissions (Delkash et al., 2016).
The temperature and pressure data were measured by a handheld meteorological instrument
(HWS1000, ZOGLAB, China) with a temporal frequency of 1min. The boundary layer data were
obtained from https://zenodo.org/records/6498004 (Guo et al., 2022) based on a merging of reanalysis
data with observations (Guo et al., 2024).

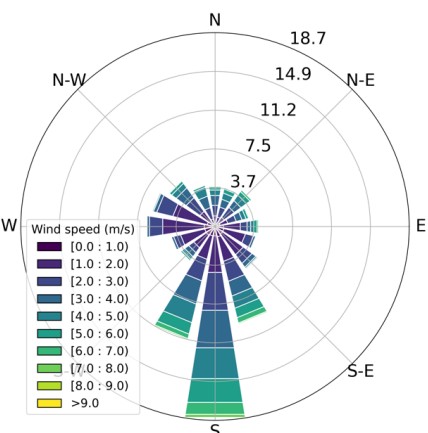


**Figure 4. The wind rose of all observed wind speeds from August 10, 2022 to August 25, 2022.**

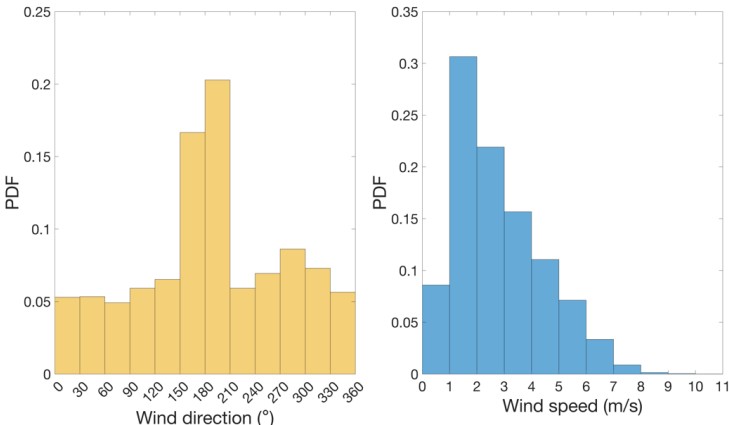


**Figure 5. Probability density function (PDF) for all observed wind direction (yellow) and wind speed (blue)**
**from August 10, 2022 to August 25, 2022.**
**2.4 Quantitative Estimation of CH$_4$ Emissions**
A mass conserving approach was used to estimate the CH$_4$ emissions in connection with the high
frequency observations of CH$_4$ and meteorological data, hereafter called the Mass Conserving Model of
Measured CMM (MCM$^2$). This approach is based on previous dynamic emissions estimates of
tropospheric atmospheric column observations of short-lived NO$_2$ (Li et al., 2023; Qin et al., 2023b), but
has never been applied to surface observations in general, or CH$_4$ in specific. Adopting this approach to
solve for CH$_4$ is done starting with the continuity equation for the conservation of mass (Equation 1),
reorganizing the individual terms and converting coordinates from space to time (Equation 2) and finally
combining the terms (Equation 3) as follows:
$$\frac{\partial CH_4}{\partial t} = E_{CH_4} - \nabla(U \times CH_4) \qquad (1)$$

$$\nabla(U \times CH_4) = CH_4 \times \nabla U + U \times \nabla CH_4 = \alpha \times \left(CH_4 \times \frac{\partial U}{\partial t} + U \times \frac{\partial CH_4}{\partial t}\right) \qquad (2)$$



$$\frac{\partial CH_4}{\partial t} = E_{CH_4} - \alpha \times \left( CH_4 \times \frac{\partial U}{\partial t} + U \times \frac{\partial CH_4}{\partial t} \right) \qquad (3)$$

where $CH_4$ is the $CH_4$ concentration (ppm), t is the time (min), $E_{CH_4}$ is the $CH_4$ emissions flux (ppm min
[-1]), and U is the wind speed (m s[-1]). The $\nabla$ is a mathematical operator that takes the gradient on spatially
distributed variables. However, when considering motion along one-dimension, the relationship between
distance, speed, and time can be used to rewrite the spatial derivatives of $\nabla(CH_4)$ and $\nabla(U)$ as temporal
derivatives (Brasseur and Jacob, 2017), where $\alpha$ is a conversion coefficient between distance and wind
speed.

The gradient term used in these equations take into account the local topography of Shanxi, which

is known for its significant features and surrounding mountains. These geographical features can impact
the transport and dispersion of $CH_4$, and their effects are incorporated into the wind field in the continuity
equation. Notably, when dealing with a non-divergent wind field, the gradient term simplifies to the term
$(U \times \nabla CH_4)$ (Sun, 2022). Uncertainty analysis was conducted before calculating the $CH_4$ emissions to
ensure only reliable data was used, since observed variation of $CH_4$ over time is influenced not only by
$CH_4$ emissions, but also changes in wind speed and pressure. Specifically, $CH_4 \times \frac{\partial U}{\partial t}$ represents the
change in $CH_4$ influenced by pressure, while $U \times \frac{\partial CH_4}{\partial t}$ represents the change in $CH_4$ influenced by
advection. Furthermore, since there is uncertainty in the observations, this work takes a conservative
approach, and only considers data when the threshold given by equation (4) is observed to be considered
influenced by emissions ( a lower threshold can be selected like 25% or 15% et al., but uncertainty will
increase).
$$u \times \frac{\partial CH_4}{\partial t} / \nabla(U \times CH_4) > 30\% \qquad (4)$$

The remaining data (approximately 22%, presented as red circle indicators in Figure 6) is not processed
in the emissions calculation as the signal is most likely due to a combination of observational uncertainty
and white noise (Prinn et al., 1987).



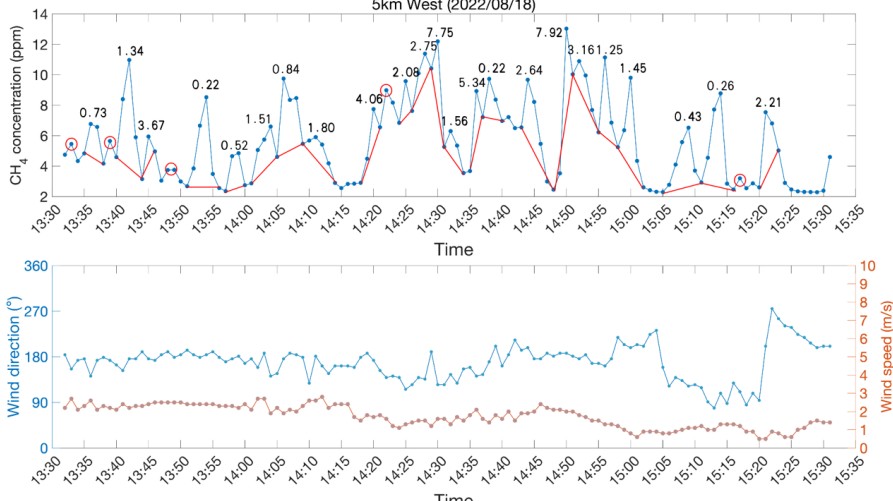

**Figure 6. Time series of CH₄ concentrations (top, blue), background concentrations (top, red), wind direction (bottom, blue) and wind speed (bottom, red) measured 5km west of CM-A on August 18, 2022. MCM² computed CH₄ emissions (top, numbers) (ppm min⁻¹) are computed for all regions where the observations are enhanced compared with the background for at least 3 consecutive observations, and which further pass the noise threshold (Equation 4).**

**2.5 Uncertainty Analysis**

In order to reduce the uncertainty of the CH₄ emission estimation, only data above the threshold given by equation (4) is consided. Prior to this, uncertainty analysis was also conducted on the relevant variables in actual experiments. As shown in Figure 7, a 5% uncertainty was assigned to both the CH₄ concentration and wind speed data, and the CH₄ emission flux was calculated. The uncertainty analysis results indicate that probability distribution of all possible calculated emissions are consistent, and the errors are smaller than 5% in each case, consistent with Equation 3 leading to a dampening of the uncertainty, as also observed in Qin et al. (2023a) study. Therefore, we believe that the results of CH₄ emissions in this study can be trusted.





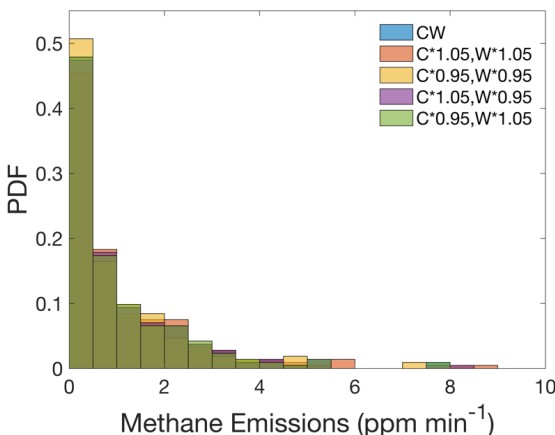


**Figure 7. The PDFs of uncertain analysis results (C represents concentration, W represents wind speed).**


**2.6 Attribution Analysis**
A 2-box mass conserving model (based on equation 5) was used to attribute CH₄ emissions from the
more than one suspected source of CH₄ in the 5k west. The changes in CH₄ over time t (min) at the
observation point $C_{coal\ mine}$ (ppm) is driven by emissions from the upwind coal mine $E_{coal\ mine}$ (ppm
min⁻¹) and the concentration gradient computed using the wind U (m s⁻¹), and the background
concentration  $C_{background}$ (ppm) as demonstrated in Figure 8.
$$\frac{\partial C_{coal\ mine}}{\partial t} = E_{coal\ mine} + U \times C_{background} - U \times C_{coal\ mine} \qquad (5)$$
All observed data points and computed emissions are used when wind direction is capable of
transporting the CH₄ from either CM-A or CM-B towards the observation site (Figure 8), while the
remaining data is not used. A discretized version of Equation 5 is given in Equation 6 and solved using a
first order finite difference approach:
$$C_{Coal\ mine_{\tau i+1}} - C_{coal\ mine_{\tau i}} = E_{coal\ mine_{\tau i}} + U_{\tau i} \times C_{background_{\tau i}} - U_{\tau i} \times C_{coal\ mine_{\tau i}} \qquad (6)$$
where $\tau_i$ and $\tau_{i+1}$ are the current and next time step, and the other terms are defined as in equation 5.
All possible sets of steady-state concentrations are computed using all possible combinations of
emissions and concentrations as boundary and initial conditions and running the equation forward to
equilibrium. The computed concentrations are analyzed probabilistically by comparing the modeled CH₄
probability density function (PDF) with the observed CH₄ PDF. Differences between the PDFs are clearly
associated with the different wind directions and hence geophysical locations of the sources can be
distinguished.



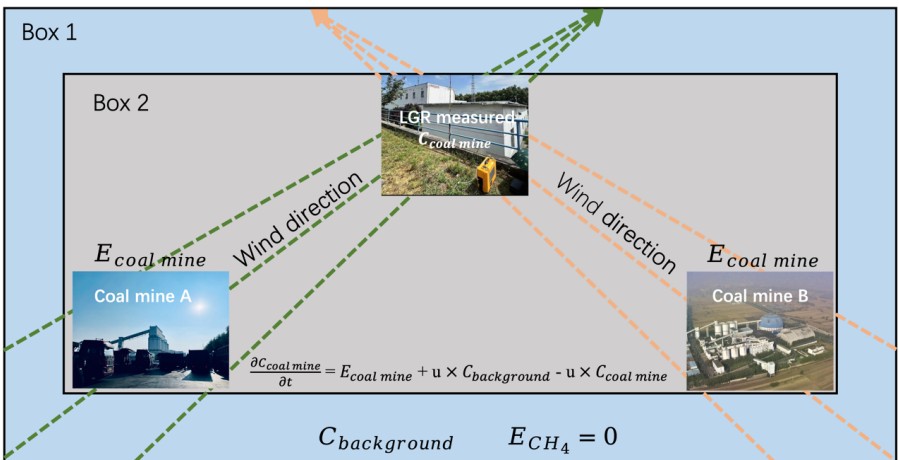


**Figure 8. Schematic diagram of the 2-Box model.**

**2.7. Converting Emissions into Policy-Relevant Units**

In order to compare the emissions with some other studies, the units (ppm min$^{-1}$) were converted

into policy-relevant units (kg h$^{-1}$), although as outlined below, this conversion leads to a larger uncertainty
range. According to the attribution analysis in Section 2.6, when the wind direction is located within a
60° arc of coal mine A or coal mine B (Figure 8), the respective CH$_4$ emissions which successfully passed
attribution were assigned to the respective coal mine. Therefore, based on the wind direction, CH$_4$
emissions of coal mine A were screened from the CH$_4$ emissions captured at North 1km, and CH$_4$
emissions of coal mine B were screened from the CH$_4$ emissions captured at West 5km. adopting the
following equation (7) to converte the units from ppm min$^{-1}$ to kg h$^{-1}$:
$$E'_{CH_4} = E_{CH_4} \times \rho_{air} \times H \times A \times 60 \quad (7)$$
$$\rho_{air} = \frac{P \times M_{air}}{R \times T} \quad (8)$$
where $E'_{CH_4}$ is the CH$_4$ emissions (kg h$^{-1}$), $E_{CH_4}$ is the CH$_4$ emissions (ppm min$^{-1}$), $\rho_{air}$ is the dry gas
density (kg m$^3$) (based on equation 8), H is the height of the vertical rise that the emissions undergo
within their first minute (m), A is the area (m$^2$) swept over an arc, which ranges linearly from 60° under
slow wind conditions to 30° over very fast wind conditions, based on the wind speed when the direction
is found to lead to successful attribution, P is the atmosphere pressure (Pa) over the sampling duration,
M$_{air}$ is the molecular weight of dry air, which is a fixed constant (28.97×10$^{-3}$ kg mol$^{-1}$), R is the universal
gas constant (8.314J mol$^{-1}$ k$^{-1}$), T is the air temperature (K) over the sampling duration.

Two different assumptions are made for the vertical extent of the plume rise, since the emissions are

computed minute-by-minute which is shorter than the adjustment time throughout the entire boundary
layer (Vaughn et al., 2018; Zinchenko et al., 2002). The first is to assume it has mixed within the bottom
one fourth of the boundary layer, and the second is that it has mixed based on a steady vertical rise equal
to one tenth of the horizontal wind. In this work, results using both assumptions will be presented.
**3. Results and Discussion**



**3.1 Spatial Distribution Characteristics of CH₄ Concentration Around Coal Mine**

Time series of $CH_4$ concentration, wind speed, and direction at 1 km, 3 km and 5 km north of CM-A are given in Figure 9a and Figure 9b. The wind direction blew from CM-A towards the observations (between 150° and 210°) 59.2% of the time, with only one day observed at 1km north (August 15) with a significant amount of wind from the west (between 240° and 300°) 92.8% of the time. Consistent with CM-A being the major source at 1km, when the wind blew from the south, the $CH_4$ concentration (3.45±0.79 ppm) was both higher and had a larger variation than when the wind blew from the west (2.40±0.17 ppm) which was similar to background conditions. This is consistent with there being no known significant sources to the west from this observation location, as shown in Figure 2. Similarly, under faster than average wind conditions from the direction of CM-A (on August 21 the mean wind was 5.70 m s⁻¹ with 14.9% of observations faster than 7 m s⁻¹), the observed concentrations were slightly lower, yet similarly variable (3.17±0.82 ppm). All of these findings are consistent with transport dominating the concentrations at 1 km north, and that high frequency wind and concentration observations are required in tandem to compute the required spatial gradients in the $CH_4$, otherwise there is no basis to objectively separate the effects of the emitting region (CM-A) from the background.

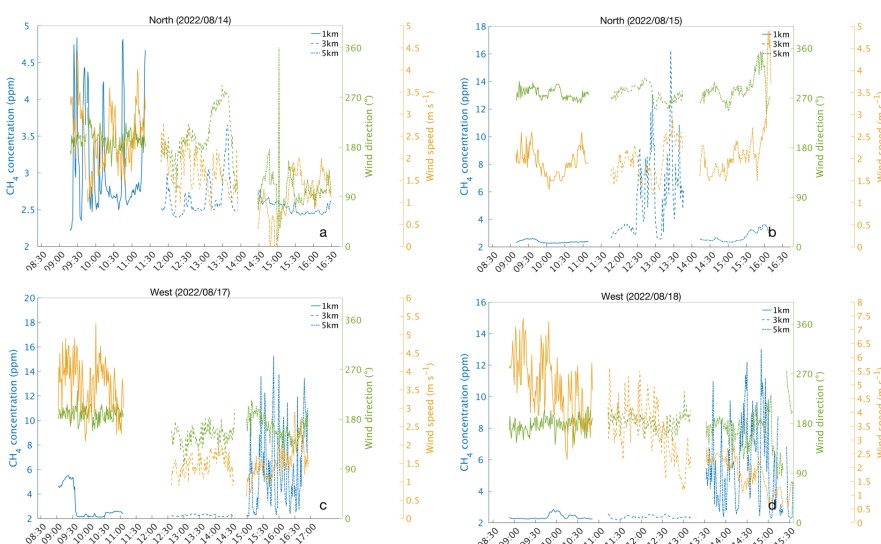

**Figure 9. Time series of CH₄ concentration (ppm) (blue), wind speed (m s⁻¹) (yellow) and wind direction (º) (orange lines) measured at 1 km (solid), 3km (dashed) and 5km (dash-dot) located north (top) and west (bottom) of CM-A on four different days.**

A similar set of findings were observed at 3 km north, while 5 km north is generally similar to the



background. At 3 km north, when the wind was from the south (59.3% of data), the concentration was
lower and more variable (3.16±1.48 ppm, with 78.7% of observations below 3.0 ppm) than at 1 km north,
consistent with advection from CM-A and a relatively stable atmosphere with a small contribution from
diffusion between the plume and the background. When the wind blew from other directions, the
distribution of concentrations broadened considerably, with a range from background (2.25 ppm) through
extremely polluted (16.2 ppm). One subset of this was observed on August 15 (observed over a total of
61 mins of observations, 6.68% of the total observations at 3 km north) when the wind was from the west
and slow, where the concentration was (5.44±2.82 ppm), as depicted in Figure 9. The data on this day
aligned with the presence of a major highway west of the observation site, which was observed in-person
to have heavy traffic consisting of vehicles carrying coal (which could still be outgassing) as well as
others powered by compressed natural gas (CNG). At 5km north the overall concentration (2.40±0.28
ppm) was generally lower than at 3km and had much lower variability, consistent with background $CH_4$.

Time series of $CH_4$ measured at 1 km, 3 km and 5 km west of CM-A and corresponding wind

direction and speed are given in Figure 9c and Figure 9d. Overall, the main wind direction is from the
south 98.4% of the time at 1km, 74.5% of the time at 3km, and 70.2% of the time at 5 km, and the wind
speed was very high when measuring $CH_4$ at 1 km west, with an average value of 4.28±1.13 m s$^{-1}$ and a
maximum of 7.4 m s$^{-1}$. This set of findings is consistent with clean upwind sources. Accordingly at 1km
west, the observed $CH_4$ concentration was slightly higher than background and had similar variability to
1km and 3km north (2.71±0.94 ppm and 86.5% of the data below 3 ppm). At 3 km west, $CH_4$ was
observed to be similar to the background (2.32±0.09 ppm). The only exception was found at 1 km west
between 9:00 am and 9:30 am on August 17, in which all of the observations were greater than 4 ppm.
Since the areas to the west from 1 km west contains mostly farmland, there was no expected strong source
of $CH_4$, as shown in Figure 2. This indicates that during this special short time, the observed slow increase
and rapid fall-off in $CH_4$ concentration must be due an unidentified source, or a change in the boundary
layer or vertical mixing structure.

Following this, it was anticipated that the 5  km west site would exhibit background types of

conditions, however the observed data deviates significantly. Wind speed was low (1.63±0.54 m s$^{-1}$,
maximum 3.0 m s$^{-1}$), $CH_4$  concentration was both very high and exhibited substantial temporal
variability (5.83±2.99 ppm, 66.7% exceeding 4 ppm, and peak of 15.3 ppm), and 70.2% of the
observations were from the south as demonstrated in Figure 10d,e,f. From Figure 1, it can be seen that



there is another coal mine (CM-B) located about 1 km away from the 5 km west measurement point, to
its southwest, although CM-B has an annual production of about 3 million tons (smaller than CM-A) and
not considered to be high gas (like CM-A), and therefore was not previously considered important. The
overlap of high concentrations with low a priori emissions, suggests that formal attribution is essential
to quantitatively confirm whether CM-B is the source responsible for both typical conditions at 5 km
west, as well as the long-range transport event at 1 km west.

$CH_4$ concentrations and wind observations in all directions except to the west, and except for the

small number of special events documents above, exhibit PDFs that show there is a decrease in
concentration the further the distance from CM-A (Figures 10 and 11), indicating that CM-A is consistent
with the major sources in these regions. These decreases are observed in terms of the median, mean,
distribution width, and percentage over 4.0 ppm all decreasing from 1 km north to 3 km north and again
from 3 km north to 5 km north.

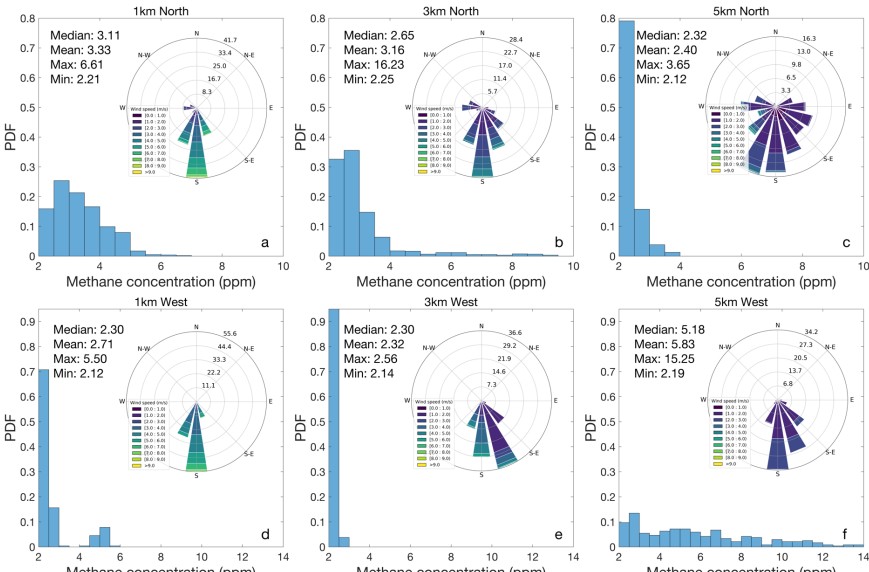


**Figure 10. Probability density map for CH₄ concentration and wind rose measured at 1 km north (top left),**
**1 km west (bottom left), 3 km north (top center), 3 km west (bottom center), 5 km north (top right), and 5 km**
**west (bottom right) of CM-A.**



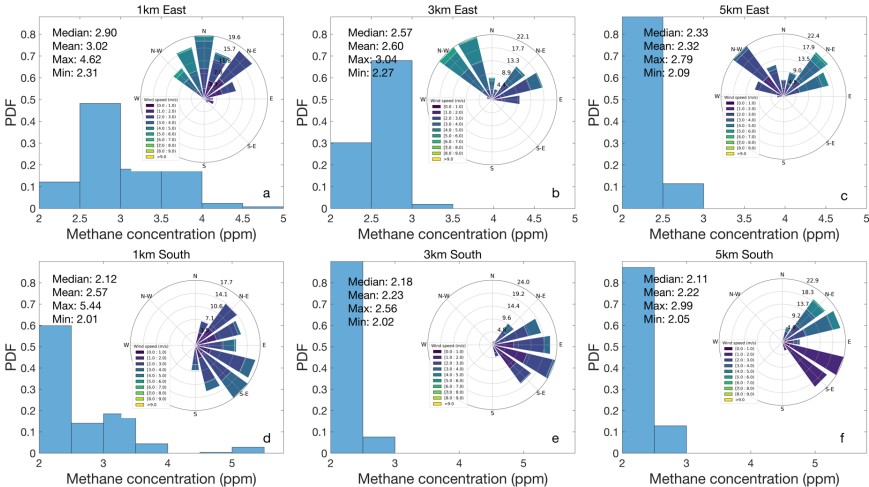

**Figure 11. Probability density map for CH₄ concentration measured at 1 km, 3 km and 5 km east and south of CM-A and corresponding wind rose chart.**

The observed CH₄ concentration gradient as one moves westward from CM-A is inconsistent with the other ordinal directions (Figure 10d, e, f). While there was a small decrease in the mean and distribution breadth from 1 km west to 3 km west, there was a large increase in the median, mean, distribution width, percentatge over 4.0 ppm from 3 km west to 5 km west. Furthermore, the data at 5 km west was found to be skewed differently than at the other sites, with approximately 70% of the data greater than 4.0 ppm. The data clearly indicates that the 5 km west site behaves more like a source region than even the 1 km north site.

**3.2 Quantification and Emission Characteristics of CMM**

The CH₄ emissions have been computed at each of the observation poits, with 25.7% of observations yielding emissions results. The PDFs of the CH₄ emissions (Figures 12 and 13) reveal that the three stations in the north and the 5km west station all are relatively high and variable, while the remainder are relatively low and non-variable. Among all the CH₄ emissions results, the highest median, mean, maximum, and breadth of the distribution are all observed at 5km west. The 3km south location has the lowest emissions of all points observed (by median), with a respective median, mean, maximum, and percentage greater than 1.0ppm/min of (0.03 ppm min⁻¹, 0.26 ppm min⁻¹, 0.90 ppm min⁻¹, 0%) (Figure 13), and is subsequently considered representative of background emissions in this work. It is important to note that there is no area within this region that has 0ppm/min emissions and that the minimum concentration on average is about 2.23 ppm (Figure 11), both of which are considered very high or



polluted compared with most other current studies (Irakulis-Loitxate et al., 2021; Sadavarte et al., 2021).

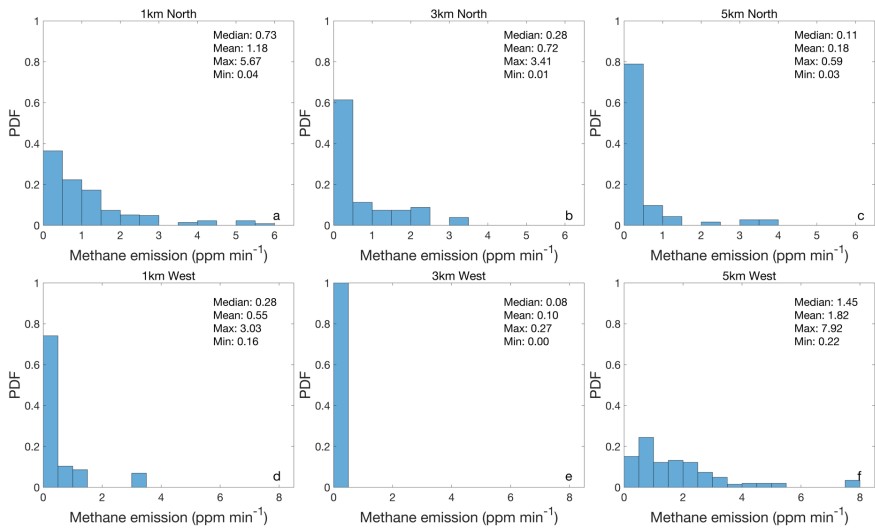

**Figure 12. Probability density functions (PDF) of computed CH₄ emissions located at 1 km north (a), 3 km**
**north (b), 5 km north (c), 1 km west (d), 3 km west (e), and 5 km west (f) of CM-A, including median, mean,**
**maximum, and minimum statistics.**

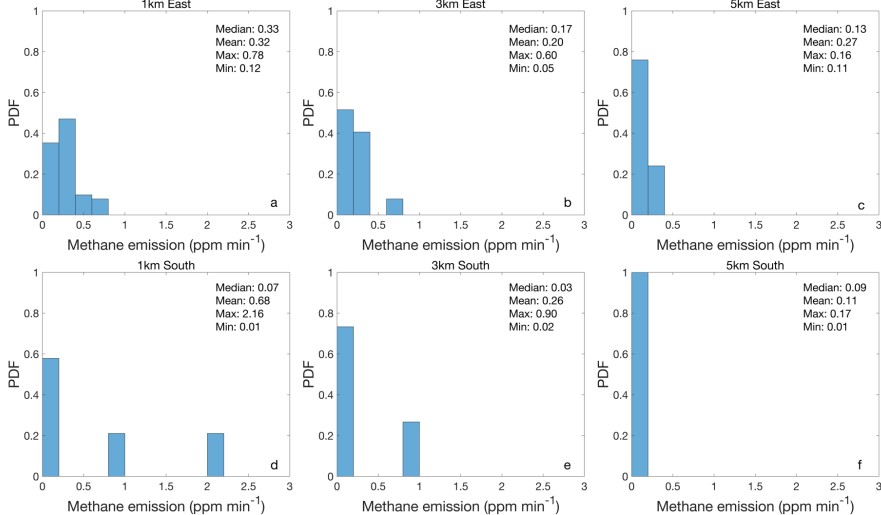

**Figure 13. Probability density functions (PDF) of computed CH₄ emissions located at 1 km east (a), 3 km east**
**(b), 5 km east (c), 1 km south (d), 3 km south (e), and 5 km south (f) of CM-A, including median, mean,**
**maximum, and minimum statistics.**

The spatial distribution Characteristics of the CH₄ emissions is similar to that of the CH₄

concentration observations (Figure 12). First, there is a decrease as one moves northward along the axis
away from CM-A, with the median, mean, maximum, and percentage of emissions greater than 1.0 ppm



min$^{-1}$ at 1 km north (0.73 ppm min$^{-1}$, 1.18 ppm min$^{-1}$, 5.67 ppm min$^{-1}$, and 42%) all larger than at 3km
north (0.28 ppm min$^{-1}$, 0.72 ppm min$^{-1}$, 3.41 ppm min$^{-1}$, and 29%). The values at 3 km north are also
larger than those at 5 km north, which respectively are 0.11 ppm min$^{-1}$, 0.18 ppm min$^{-1}$, and 0.59 ppm
min$^{-1}$, and 0%. The subset of emissions under low wind speed conditions exhibited a larger decline from
1 km to 3 km and from 3 km to 5 km. The observations are further consistent with transport from a single
dominant source located at CM-A being the primary driving factor, and diffusion from other industrial
sources in Changzhi city center being a secondary factor.
Consistent with there being few to no sources impacting the 1 km west and 3 km west sites, except
for considerably less transport from CM-A the computed PDFs at these sites (Figure 12) demonstrate
low emissions and low variability, with the respective median, mean, maximum, and percentage of
emissions greater than 1.0 ppm min$^{-1}$ at 1 km west being 0.28 ppm min$^{-1}$, 0.55 ppm min$^{-1}$, 3.03 ppm
min$^{-1}$, and 16% and at 3km west being even lower (0.08 ppm min$^{-1}$, 0.10 ppm min$^{-1}$, 0.27 ppm min$^{-1}$, and
0%). However, the $CH_4$ emissions computed at 5 km west were the highest and most variable of all results
computed in this work, with the respective statistics being 1.45 ppm min$^{-1}$, 1.82 ppm min$^{-1}$, 7.92 ppm
min$^{-1}$, and 60%. Furthermore, the skewness of the distribution at 5 km west (which has 30% of the $CH_4$
emissions above 2.0 ppm min$^{-1}$) is much larger than at 1 km north (which only has 15% of emissions
above 2.0 ppm min$^{-1}$). Combining these pieces of information, at first look it seems that the site at 5 km
west is not related to the emissions from CM-A, or at best are a mixture of emissions from CM-A and
those at another site, herein proposed to be CM-B. The remainder of this study focuses on disentangling
and attributing contributions from CM-A and CM-B at 5km west, with the observations at the remaining
sites ruled out in terms of having a contribution from CM-B.
**3.3 Attribution of $CH_4$ Emissions**
This works applied the 2-box model at the 5 km west site and quantified the contribution of both
CM-A and CM-B emissions to the observed $CH_4$ concentration distributions as given in Figure 14. First,
the results of the 2-box model produce PDFs which overlap with the overall observed $CH_4$ PDF,
indicating that the results are reasonable. Second, space of the emissions computed from the two different
two coal mines do not overlap, and cover two independent portions of the observed $CH_4$ PDF. Specifically,
the 30%, 50%, and 70% values of $CH_4$ concentration observed at 5km west are 3.68 ppm, 5.18 ppm, and
6.86 ppm respectively. The emissions from CM-A yield a $CH_4$ concentration less than 4 ppm most of
the time, with a 30%, 50%, 70%, and maximum concentration of 2.96 ppm, 3.15 ppm, 3.31 ppm, and





4.60 ppm, while the emissions from CM-B yield a CH$_4$ concentration more than 5 ppm most of the time,
with a minimum, 30%, 50%, 70%, and maximum concentration of 4.76 ppm, 5.20 ppm, 5.68 ppm, and
6.18 ppm.

Overall, the emissions from CM-B cover well the observed concentration values from the range of

50% to 70%, with a single high value around the 90% value, while the emissions from CM-A cover well
the observed concentration values in the range from 10% to 30%. One weakness is that the length of
observations is not as comprehensive as at the other sites, and therefore it is possible that had more
observations been made, the contributions from CM-B would have filled more of the space between the
70% and 90% levels, and some combination of sources from CM-A and CM-B would have better filled
the space between the 30% and 50% levels. The results indicate to a high degree of certainty that the
emissions from the two respective coal mines are distinct, with CM-A the source of emissions in the
lower range of the concentration distribution and CM-B the source for emissions in the higher
concentration range, covering values in the middle and upper range. Improvements in modeling,
additional observations, considering possible contributions from additional missing sources, and
consideration of longer-range transport could add further improvement and better explore the
intermediate range of observed concentrations.

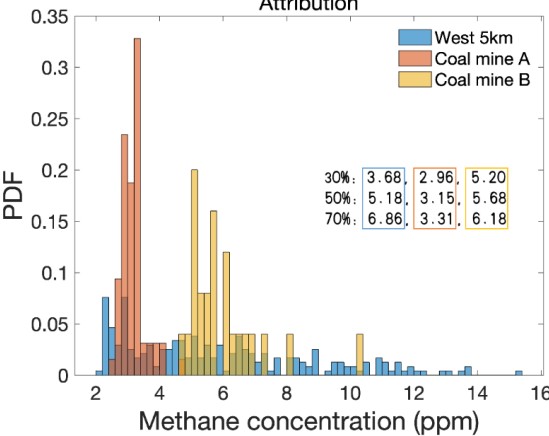


**Figure 14. The PDFs of CH$_4$ concentration measured at 5 km west (blue) and simulated using the 2-Box model**
**under conditions when the source is CM-A (red), and when the sources is CM-B (orange), including**
**representative 30%, 50%, and 70% bounds are in.**
**3.4 Policy-Relevant Emissions**

In order to compare the values of CH$_4$ emissions from the Shanxi coal mines computed in section



2.7, the units (ppm min$^{-1}$) are transformed into units of (kg h$^{-1}$) via a conversion factor based on equations
(7) and (8). This conversion increases the overall uncertainty, since since it involves approximations of
the area swept, the boundary layer height, and and other uncertainties. In this study, average CH$_4$
emissions from CM-A and CM-B are 13670±7400 kg h$^{-1}$ and 5070±2270 kg h$^{-1}$, and the CH$_4$ emissions
range is from 200 kg hr$^{-1}$ to 67700 kg hr$^{-1}$ and 430 kg hr$^{-1}$ to 15300 kg hr$^{-1}$ repectivily (Table 2). Both mines
display a fat tail distribution, with respective 25$^{th}$, median, and 75$^{th}$ percentile values of [870, 7500, 38700]
kg h$^{-1}$ and [431, 1590, 7000] kg h$^{-1}$. respectively. These findings are demonstrated to be higher than CH$_4$
emissions from equivalent oil and gas operations in the USA (Chen et al., 2022), with one site being
roughly double and the other similar to and slightly lower than day-to-day emissions inverted over 5-
years from TROPOMI (Hu et al., 2024) over the same region. This is consistent with the fact that the
results herein target very high frequency and spatially confined emissions, while satellites provide day-
to-day values over a larger pixel area, as well as associated significant uncertainties involved in
converting from ppm to kg. In specific, at CM-A, the minimum (200 kg h$^{-1}$) and maxmum (67700 kg h$^{-1}$
) values of CH$_4$ emissions results are larger than the respective minimum (8 kg h$^{-1}$) and  maxmum (37300
kg h$^{-1}$) values of CH$_4$ emissions inverted from TROPOMI, with the statistical values roughly double,
while at CM-B, the minimum (430 kg h$^{-1}$) value of CH$_4$ emissions results is greater than the minimum
(20 kg h$^{-1}$) from TROPOMI, the maxmum value  (15300 kg h$^{-1}$) of CH$_4$ emissions is less than the
maximum (37300 kg h$^{-1}$) from TROPOMI, and the statistical values are slightly smaller although similar.
In this study, observations were made within 1km of the coal mines on a minute-to-minute basis, while
the  the TROPOMI observed the xCH$_4$ over a space scale (5.5×7 km$^2$) and on a day-to-day average basis,
both of which indicate advantages of higher sampling diversity, especially so at the extreme values
observed, when compared with TROPOMI's results. For this reason, it is likely that the sampling time at
CM-B was insufficient to fully capture the fat tail of the emissions, since the maxmum is smaller than
the maximum estimated CH$_4$ from TROPOMI (consistent with the limited two days of data at coal mine
B), (Table 1). Given that the emissions are generally larger at the higher production coal mine, they are
consistent with the concept that over geologically similar environments, higher coal mine production
leads to increased CH$_4$ emissions, although the increase is not linear as most current models assume.









**Table 2. The CH$_4$ emissions (kg h$^{-1}$) of CM-A and CM-B using different observation methods and statistical**
**methods**

| Coal mines | High frequency ground observation CH$_4$ emissions (kg h$^{-1}$) | | | | | | | | TROPOMI inverted CH$_4$ emissions (kg h$^{-1}$) | | | |
|---|---|---|---|---|---|---|---|---|---|---|---|---|
| | 1/4 boundary layer | | | | 1/10 horizontal wind | | | | | | | |
| | Mean±SE | Min | Median | Max | Mean±SE | Min | Median | Max | Mean±SE | Min | Median | Max |
| **CM-A** | 13670±7400 | 200 | 7500 | 67700 | 19100±9800 | 70 | 5790 | 63800 | 5500±700 | 8 | 2130 | 24900 |
| **CM-B** | 5070±2270 | 430 | 6060 | 15300 | 1000±444 | 30 | 1600 | 2850 | 6200±1000 | 20 | 1450 | 37300 |

**4. Conclusions**
This study presents a high frequency ground observation campaign and a new analytical top down
emissions estimation approach to quantify the emissions of CH$_4$ from a high gas coal mine region with
multiple mines. The base observations are made using a portable greenhouse gas analyzer in connection
with meterological and other optical measurements. Observations have been made over 15 days at a
frequency of 1 second, at various locations of known distance from an existing high production coal
mine. The high frequency observations are then used in connection with a mass conserving modeling
platform to estimate the CH$_4$ emission rate. A mass-conserving Two-Box model was used for attribution
analysis in this study. The results show that the spatial characteristics of CH$_4$ concentration/emissions are
consistent with the distance from a well characterized of single coal mine within 5km distance, and CH$_4$
emissions demonstrate clear first order effects of both transport and diffusion, with methane emission
rates of 0.73, 0.28 and 0.15 ppm min$^{-1}$ at 1, 3 and 5 km downwind respectively. At 5 km north the overall
concentration (2.40±0.28 ppm) was generally lower than at 3km and had much lower variability,
consistent with background CH$_4$, which demonstrate that the CMM emissions mainl affect the
surrounding area with 5km distance. However, the overlap of two coal mines (CM-A and CM-B) have a
far more complex distribution of emissions intensity, ranging as high as 1.82 ppm min$^{-1}$, which is much
higher than the emissions of single source at any directions. Another, the background concentration of
surface CH$_4$ in the mining areas is very high compared with other studies, with a value always at and
above 2.23 ppm. Finally, in order to compare these results with results from other parts of the world, the
subset of emissions which successfully underwent attribution were converted into the unit of kg h$^{-1}$ using
an approximation of the volume swept by the wind and other approximations of the atmosphere. The
resulting values were found to be  13670±7400 kg h$^{-1}$ and 5070±2270 kg h$^{-1}$ respectively, which are higher



than $CH_4$ emissions from equivalent oil and gas operations in the USA, and in one case are higher than
but in the other case similar to day-to-day emissions inverted from 5-years of TROPOMI over the same
region.

This work demonstrates that high frequency surface observations of $CH_4$, in combination with high

frequency observations of wind can provide deep insights into emissions by accounting for high
frequency changes in space and time at the same time, which tend to be missing from models which used
more idealized approaches (such as average plume shapes and sizes, levels of coal production, and
interpreting gradients from a small number of fixed images). A significant source of $CH_4$ emissions from
a previously unknown or improperly classified mine may pose a vastly different range of observed
concentration as well as computed emissions than expected. The importance of observations at both high
frequency and regional spatial coverage are demonstrated, and a set of practical methods that are freely
open and can be adopted and modified rapidly are provided. The approach to source attribution used
herein can provide insights to policymakers to formulate regional emission control policies and provide
a check on or a priori assumption for the new generation of advance satellite-based top-down emissions
estimates, while demonstrating that spatial attribution is a critical next-step for satellite approximations
and $CH_4$ control policies.
**Data Availability**
All underlying data herein are available for access by the editors and reviewers at
https://figshare.com/s/1a393772d7b72ae17e62 and will be made available to the community upon
publication.
**Author contributions**
K.Q., J.B.C. and F.L. designed the research; F.L., C.Y., and Y.S. collected the data; J.B.C. and F.L.
analyzed the data; Q.T., W. H. and Q.X. provided the support for data analysis and drawing; Q.H., S.W
gave suggestion on running the 2-Box model; F.L. wrote the manuscript with inputs from J.B.C., Q.H.
and P.T.; All authors discussed the results and contributed to the final manuscript.
**Competing interests**
The authors declare that they have no known competing financial interests or personal relationships that
could have appeared to influence the work reported in this paper.
**Disclaimer**
Publisher's note: Copernicus Publications remains neutral with regard to jurisdictional claims made in



the text, pub- lished maps, institutional affiliations, or any other geographical rep- resentation in this
paper. While Copernicus Publications makes ev- ery effort to include appropriate place names, the final
responsibility lies with the authors.
**Acknowledgments**
We sincerely appreciate all the scientists, engineers, and students who participated in the field campaigns,
maintained the measurement instruments, and helped with and collection and processing of the data.
**Finacial support**
This study was funded by the National Natural Science Foundation of China (42075147, 42375125) and
the Fundamental Research Funds for the Central Universities (2023KYJD1003, JP230021).

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
