# Peer review of "Surface Observation Constrained High Frequency Coal Mine"

_EGUsphere, 2024_

## Author Response (AR1)

Dear Editors and Reviewers,

Thank you for your kind energy, time, and patience. We are providing a point-by-point reply to the reviewer comments below. In each case, all reviewer comments are copied and pasted below unedited in normal text, our responses are given in blue text, and corresponding changes in the manuscript are made in green text. We have worked diligently to respond to each and every comment, including: new sensitivity runs, additional analysis, production of new figures and tables, and increase of the supplemental information. In the end, the comments and reviews have helped us to produce an even stronger result, one which is clearly new, unique, and consistent. We understand that there is a lot of discussion herein. We believe that this is important, since the tradiational approaches, models, and assumptions just do not yield reasonable results in our areas observed. However, we believe that our new approach can explain a story which is consistent, scientific, and reasonable. We hope that this can be the starting point for future improvement in other areas with challenging geology, geography, intense sources, and very high air pollution levels. Thanks again to everyone for your patience, insights, and support. Looking forward to continued interaction and the production of an excellent and useful scientific product, with real world applications and demand.

Regards,

Jason Blake Cohen

jasonbc@alum.mit.edu

on behalf of the authors

**Reviewer #1:**

The reviewer finds the manuscript hard to read and understand, due to the confusion in the terms used, the vague model description for emission estimation, and poor English in the manuscript.

**1. Confusing terms used in the manuscript.**

A) $CH_4$

In equations (1), (2) and (3), $CH_4$ represents methane concentration with the unit of ppm, while the in the main part of the manuscript, $CH_4$ just means methane in English. This creates unnecessary confusion for readers.

Thank you very much for kind reminder. We have revised all the "$CH_4$" according to your suggestion and now differentiate between methane concentration (hereafter delineated as $[CH_4]$), methane emissions (hereafter delineated as ECH4), and methane as a word (hereafter is delineated as $CH_4$).

B) $CH_4$ emissions

Why one term "$CH_4$ emissions" is assigned to two different variables, one in the unit of ppm/min ($ECH_4$), and another one in the unit of kg/h ($E'CH_4$), and $ECH_4$ has never been clearly defined in the manuscript: is it the methane concentration change over time at the certain point? Or average methane concentration changer over time over a controlled volume? This is the first time that the reviewer has seen such a term is used to define emissions. Please define parameters with their actual physical meaning.

Thank you very much for this question, which can help us make better improvements to this article. In fact, ECH4 is clearly defined as methane emissions in the manuscript, with the unit being ppm $min^{-1}$, which was obtained using the mass conservation model. However, because there are new papers over the past few years which are now using the unit of kg $hour^{-1}$, in order to allow for more simple comparison, we have added Equation 7 and subsequent analysis of our results additionally using this unit. Note that adopting this unit has required additional assumptions to be made connecting our observed wind speed with an assumed area and height, which then leads to an increase in the uncertainty of the overall result. For this reason, we want to provide both sets of emissions data. In all cases, emissions of $CH_4$ are now clearly delineated as ECH4 with the respective units determining whether or not the additional equation 7 is applied.

C) Temporal frequency

Why a frequency has a unit of minute? Should it be hertz (event per time) (Lines 136, 143)

The observational frequency of the raw data from the machine used herein is 1Hz, which means one observation is obtained per second, introduced in line 107 of the manuscript (https://www.merriam-webster.com/dictionary/hertz). In order to apply the mass conservation model to calculate methane emissions, the temporal derivative of methane concentration, wind speed, and direction data need to match. To ensure this is the case, the temporal concentration data was averaged to per minute, as mentioned in line 113. Hereafter, the unit of per minute is used.

Line 107: "Atmospheric $CH_4$ concentrations at 5 m above the surface were observed daily at 1 Hz from 8:30 to 17:00 local time in August 2022 using two portable greenhouse gas analyzers (LGR-915-0011, California, USA)."

Line 113: "The $CH_4$ concentrations data was averaged minute-by-minute to match observed wind data, and subsequently used to compute $CH_4$ emissions."

**2. The Mass Conserving Model of Measured CMM and the 2-Box model**

First, the model or models used to estimate methane emissions from one or multiples coal mines are not well described. The reviewer is expecting the following information to be clearly stated in the model:

A) Is the model dealing with one coal mine or multiple coal mines?

In this study, the Mass Conserving Model of Measured CMM is solving the mass conservation equation to derive the transported emissions observed at the place and time where the observations are taken. The number or locations of the sources are not relevant at this point. However, the Mass Conserving Model of Measured CMM is used to calculate methane emissions within a certain spatial area. The methane emissions measured in this spatial area may come from a fixed methane emission source (a coal mine), from two or more methane emission sources (multiple coal mines), or even other possible sources transported from far away or due to extreme and infrequent events. For example, there is a specific case given on line 305-310 in which the most reasonable explanation for an extreme computed emissions in terms of large magnitude and infrequent occurrence were likely the result of a leaky vehicle. Due to this reason, this specific extreme emission value was not subsequently used in the attribution steps.

The application of the subsequent 2-Box model allows attribution to be obtained based on the sources being one of two different adjacent coal mines. This is done using a statistical forward and backward analysis using all possible combinations of computed emissions and wind directions and speeds which are applicable to the known directions of the two adjacent coal mines. If there are multiple methane emission sources upwind occurring at the same place and time which were transported over the sources with the same time and direction, the 2-Box model may not be able to differentiate this. The model does however produce a statistical result of concentrations obtained using the computed methane emissions from each emission source. The distributions of

computed concentrations are shown to be consistent with the observed distributions of measured concentrations, allowing confidence in the attribution.

B) Is the coal mine emission treated as point source or area source?

In reality, there is no difference between the two, since all sources are area sources. It is just that point sources are emitted over a very small area. In this work we are treating the emissions using a Lagrangian parcel when computing the MCM$^2$ approach. We then switch to a specific set of assumptions of a box in space and time when converting units to kg h$^{-1}$.

C) Are the methane emissions from the coal mine considered stable or not?

Our computed emissions range in time from a minimum of 3 minutes to a maximum of 22 minutes for each event. There is not any single observed event which is sufficiently long to be stable at the per hour time scale. This is clear from the variation in the observations, and is the underlying reason why this work has focused on updating the mass balance equation at high-frequency. Gaps and differences in terms of magnitude and timing are presented in Figure 5 and in the underlying dataset accompanying the paper https://figshare.com/s/1a393772d7b72ae17e62.

[Figure]

**Figure 5. Time series of CH$_4$ concentrations (top, blue), background concentrations (top, red), wind direction (bottom, blue) and wind speed (bottom, red) measured 5km west of CM-A on August 18, 2022. MCM$^2$ computed CH$_4$ emissions (top, numbers) (ppm min$^{-1}$) are computed for all regions where the observations are enhanced compared with the background for at least 3 consecutive observations, and which further pass the noise threshold (Equation 4).**

D) What is the control volume that the model is applying?

The concept of a control volume is only applied when converting the emissions results from ppm/min to kg/hr. This is detailed in the manuscript in equations (7) and (8) and section 2.7. In specific, it assumes a plume rise height and area based on wind speed, boundary layer height, and angle of spread (itself a function of the wind speed), thus obtaining a hypothetical control volume. The point is that the plume cannot flow any higher or further than the wind speed allows it to be transported over the length of the total emissions event. The manuscript states the following:

"H is the height of the vertical rise that the emissions undergo within their first minute (m), A is the area ($m^2$) swept over an arc, which ranges linearly from a maximum of $60°$ under slow wind conditions to a minimum of $30°$ over very fast wind conditions, based on the wind speed when the direction is found to lead to successful attribution."

"Two different assumptions are made for the vertical extent of the plume rise, since the emissions are computed minute-by-minute which is shorter than the adjustment time throughout the entire boundary layer (Vaughn et al., 2018; Zinchenko et al., 2002). The first is to assume it has mixed within the bottom one fourth of the boundary layer, and the second is that it has mixed based on a steady vertical rise equal to one tenth of the horizontal wind. In this work, results using both assumptions will be presented."

E) What are the boundary conditions and initial conditions if transit process is considered?

The 2-box model is a set of differential equations, requiring boundary and initial conditions. All possible combination of observed concentrations at the background locations, computed emissions and observed wind-fields which correspond to the appropriate physical direction of the two upwind mines are used to drive the model. All combinations are run until the values converge on a concentration. This distribution of results is then presented probabilistically. The details can be found in section 2.6 and equation (5) and (6), and the results are given in Figure 16 as well as in the dataset at https://figshare.com/s/1a393772d7b72ae17e62.

[Figure]

**Figure 16. The PDFs of CH$_4$ concentrations measured at 5 km west (blue) and simulated using the 2-Box model under conditions when the source is CM-A (red), and when the sources is CM-B (orange), including representative 30%, 50%, and 70% bounds are in.**

A scientific description of a mechanism physical transport model should include the following contents:

A) Control volume: the physical region where the model is applying. In the case of this paper, are we considering 3-dimensional box covering multiple coal mines and monitoring points with a height of boundary layer?   An illustrative figure will help the readers and authors too.

Thank you very much for your recommendation. The underlying technical issues have been explained. We have now added the following illustrated figures in the manuscript in section 2.6.

[Figure]

**Figure 7. Overview of the MCM2 and 2-box mass conserving model used in this work.**

[Figure]

Figure 9. The control volume for CH$_4$ emissions unit conversion from ppm min$^{-1}$ to kg h$^{-1}$.

B) Key assumptions of the model (3-dimension transport or one-dimension only? Transit or stable?) which will lead to

We have addressed these points above in more detail. A summary is given below.

The emissions model is a one-dimensional spatial and one-dimensional temporal model, which is not assumed to be stable in time, and is therefore computed on a minute-by-minute basis. The 2-box attribution model is assumed to be 1-dimension in space, and is solved to equilibrium in time using all possible combinations of observed emissions, wind speed, background concentration and direction.

The final conversion to the units of kg h$^{-1}$ is assumed to be 1-dimensional in time with transport filling a control volume, computed on a minute-by-minute basis, and then aggregated to per hour.

Thank you very much for your asking us to be clear on these points.

C) Control equations base on mass balance and simplified by key assumptions

Our equations fully satisfy mass balance, and are the same equations used in regional mass-balance models such as WRF-CHEM, and global mass-balance models such as GEOS-CHEM, CESM, etc. There are key assumptions made that the transport is one-dimensional in space and on-dimensional in time, although the equations are updated on a minute-by-minute basis. The divergence accompanying the observed minute-by-minute changes in the wind speed and direction are accounted for in this work.

D) Boundary equations and initial conditions

We believe that the model is explained in the manuscript. We have worked hard to improve our communication, readability, illustration, and more, and hope that the reviewer and the community has a clearer understanding at the present time.

We believe that some of the assumptions below stated by the reviewer indicate that the reviewer may be accustomed to a subset of mass balance models which are not comprehensive and have already made certain critical assumptions which we do not make in our work. We will take the time to slowly and carefully address each point, explaining how and where our model works to address the questions raised. In fact, our equations can be simplified into such a similar model, but we chose to not do so due to the fact that the observed wind and chemical concentrations are rapidly changing throughout the entire dataset, which disallows these assumptions to be made.

Secondly, the model itself is questionable. Since the model is not clearly defined in the manuscript, the reviewer cannot assess it accordingly. But in principle, the changes in methane concentration (ppm/min) downwind of a coal mine should be contributed to:

1) Variation of coal mine methane emission rates (not the absolute emission rate, but the changes)

This is not necessarily true. For example, you can have a case where the emissions are constant but the downwind observed methane concentration does change in time. This can happen when there is a divergence in the wind field. Please observe figure R1 below to highlight this case. In this case, the change in concentration downwind (in ppm min$^{-1}$ or ton min$^{-1}$) will change as a function of the absolute emissions rate. Based on our wind speed and direction data, we indicate clearly that this condition is observed to occur frequently in our study area.

Note that our equation includes this effect of both absolute emissions and changes in Emissions. Our equation includes both an active time derivative and spatial gradient of

the wind times the concentration. Note that the spatial gradient expands into multiple terms due to the chain rule. This allows for both the absolute values and the spatial and temporal changes in the computed emissions and the observed wind speed direction and the concentration to all matter.

Simple Examples: Absolute Emissions is Constant

[Figure]

Absolute emissions is constant, but the wind direction/speed change, the $CH_4$ concentration change can be observed

Figure R1. Example of case where emissions rate is constant, but downwind concentration rates vary over space and time, due to non-divergent wind conditions, similar to those observed in this campaign.

2) Changes in wind speed and direction (again, not the absolute value)

gain, the absolute value of wind speed and direction can be constant and can have an impact on the downwind methane concentration per minute change. This does occur under two different situations. First, when there is a time change in the emissions. However, it also does occur when there is a stable divergence anywhere in the air column above the observation site. In this case, both the emissions and the wind fields are constant, however, a vertical wind flow is induced. Similarly, a change in the vertical mixing between cleaner air above and dirtier air below, or a complete lofting of the emitted particles rapidly from the surface upward, so that they are not ever measured at the ground site. On top of this, there could be a total column change in the air pressure, especially if the observations are located near a mountainous area, as occurs in this work. These effects were observed within the area and times of the study herein. In all of this case, it is essential to know the absolute value of the wind speed and direction.

Note that our equation includes this effect of both the absolute values and changes in wind speed and direction to all be included. See above for detailed explanation.

3) Noise created by the instrument itself.

We have intentionally excluded all changes smaller than 30% to ensure that we are not considering observational error. This value could be changed or a sensitivity experiment carried out. We have decided to be very conservative in this case.

In other words, the changes in methane concentration (ppm/min) at a certain downwind point is not related to the absolute methane emission rate (kg/h) upwind, how can we estimate the methane emission rate (kg/h) from the downwind methane concentration change (ppm/min)?

Consider an extreme ideal scenario: a coal mine emits 1000 kg/hr as a points source, the wind is precisely eastward at constant speed of 1 m/s. The instrument downwind measures the true in-situ methane concentration without any noise. In this case, we will observe constant methane concentration with 0 methane concentration changes, and lead to 0 methane emission rate from the model described in the current manuscript, which does not make sense.

In fact, the extreme ideal scenario you haven assumed is not found to exist anywhere within the work done here for a 1-hour period. Since your assumption is in terms of kilograms per hour, then the wind must meet this assumption for an entire hour. Otherwise, the situation would not be accurate. Since the measurements are used over a long period of time, eventually the wind changes direction, and for this reason will be picked up as a change in time. On top of this, since the original equation is based on a gradient, adding in a second instrument or co-locating at a second location even just a few meters away would immediately pick up the gradient and register a vastly different number. Since in reality all air, even in your assumed case, as some amount of diffusion, therefore in reality the signal would be picked up. In fact, if the horizontal gradient across your assumed ideal line is really as strong as you say (1000kg h$^{-1}$on the line and 0kg/hr off the line) then the diffusion will be very large. This will be picked up by the derivative of the second order term (wind times concentration) very effectively. This theory is in fact supported by the equations used by mass flux observational systems (Wofsy et al., 1993).

In the process of observation, there were indeed situations where $CH_4$ concentration and wind speed changed very little over ten- to thirty-minute-long periods. Over these time periods, since the variation was under 30%, we decided to not compute emissions. This is an aspect of our quality control, to ensure that machine noise or measurement error do not lead to erroneous emissions estimations. All quality control steps applied before using the Mass Conserving Model to compute the $CH_4$ emissions are detailed in the manuscript in equations (4) and section 2.4.

However, we selected four different directions and multiple observation points around the coal mine for CH$_4$ concentration observation, with a high time frequency of 1 Hz.

Here is an example of some of the data we have obtained. Note that in the case below, we have only found that the variation in time is sufficient to compute emissions occurs over the times boxed in red. The other times still exhibit changes in concentration, but are not included in the emissions calculation since the respective change in wind times concentration is smaller than 30%.

[Figure]

If a model cannot deal with simple scenarios, it cannot treat the complex situation when transit process is considered.

We fully agree with you. We invite the models of the type you are referencing to use our data, and compute emissions at higher frequency. We would also like to see how the uncertainty range changes for the computed emissions. We look forward to working together more to compare and contrast different approaches under more realistic conditions. We believe that this is important, since we cannot find any case in our results in which the data is long enough as to compute emissions continuously over 1 hour. We have made a set of assumptions to transform our units into the emissions per hour basis, as described in section 2.7.

We believe that the approach in this paper offers a unique approach to compute CH$_4$ emissions at high frequency (even if it is rough and has uncertainty). We have derived and computed an unbiased way to consistently compute at high temporal frequency emissions, including variability, and successfully attributed these emissions to two different substantial coal mine based sources.

**Language and logical issues in the manuscript**

A) Lines 40, 41: $CH_4$ emission estimates are highly uncertain in both space and time (Brandt et 41 al., 2014; Saunois et al., 2020b).
When people talk about the spatial and temporal variability of oil gas emissions, they are referring to the real emissions, not the emission estimates.

Our work at emissions estimation is to quantify the actual emissions and an uncertainty bound at high frequency. We are talking about the same thing. It is the rate at which the $CH_4$ escapes into the atmosphere from the source. The units can be ppm/min, ton/hour, etc., but the actual thing is the same.

We have removed the word "highly" next to uncertain to be a little bit less strong. But the fact of the matter is that there is a lot of work we have done (and others as well) which indicate clearly that there are issues of spatial mis-match, temporal mis-match, and absolute value mis-match between various platforms and observations.

If the issue is the difference between bottom-up and top-down emissions estimates, then this is an interesting area. In reality the results from both should match each other, unless there is a fundamental mis-match in how emissions are being talked about and discussed.

B) Lines 40-45: For these reasons, new approaches to quantify, reduce uncertainty, and attribute $CH_4$ emissions are necessary and can provide support for policies aiming to control and mitigate 45 CMM (Cao, 2017).

New approaches are necessary not because the emissions change with time and location, nor the fat tail distribution. It is because we need more accurate and economical tools.

The original content of lines 40-45 in the manuscript is: "$CH_4$ emission estimates are highly uncertain in both space and time (Brandt et al., 2014; Saunois et al., 2020b). They also generally have a fat tail distribution, wherein a small number of samples have extremely large emissions that overwhelm emissions under average conditions (Duren et al., 2019; Plant et al., 2022). For these reasons, new approaches to quantify, reduce uncertainty, and attribute $CH_4$ emissions are necessary and can provide support for policies aiming to control and mitigate CMM (Cao, 2017)."

As you said, we need more accurate and economical tools. One of the reasons why is that many current tools are not capable of inverting the emissions at high frequency, or under complex meteorological conditions. However, as some recent work has demonstrated (Li et al., 2023; Qin et al., 2024; Qin et al., 2023), emissions variability in time may be a dominant factor behind annualized or other long-term emissions estimates. For this reason, quantifying changes at high temporal frequency alone directly contributes to being more accurate ant economical.

For example, coal mines, which are sources of methane emissions, do not emit the same amount of methane all times of day and all days of the year. There are impacts due to mine size, production, underground geology, possible emergency situations underground, water table, changes in energy, coal, materials, or other demand, efficiency of coal bed gas use or retrieval, and other possible factors as well. Satellite remote sensing approaches to computing methane emissions currently have lower spatial and temporal resolution compared to the high-frequency ground observation method used in this study. The transit time of the satellite is fixed every day. Clouds or aerosols vary day-to-day. Some of these factors of variability may introduce an extremely high event from being sampled, leading to an over (or under) sampling of such events. Additionally, low emissions events may not be distinguishable from background noise or other urban-based sources which have advected into the same grid. This could lead to an over (or under) sampling of such very low events. Given all of this, emissions data observed by satellites from coal mines cannot even meet a daily temporal frequency, let alone and hourly frequency. Therefore, we further need methane emission observation methods with higher temporal and spatial resolution to achieve more accurate estimation of methane emissions and help improve the results which are obtained from satellite estimations.

C) Line 54: Uncertainties are rarely assessed holistically or in detail (Cohen and Prinn, 2011; Cohen and Wang, 2014).

Be careful to make such a claim. Almost all methane measurement papers have one section addressing and reporting their measurement uncertainties.

Uncertainties consist of measurement uncertainties, model uncertainties, and parametric uncertainties. Thank you for your words of caution. We will more carefully word this part. However, we have reviewed many such methane measurement papers recently published, and find that a significant number of these papers do not address the modeling uncertainties and parametric uncertainties, and as such continue to always use said plume models which cannot address the issues observed in this work. Therefore, this work uses a newer approach which starts to account for these modeling and parametric uncertainties. Although whenever a new approach is raised, of course continuous improvement is important. We hope that this work can form a new basis point for this discussion.

Uncertainties in whether or not to consider the basic formulation of plume models as being reasonable under the observed conditions are rarely addressed holistically or in detail, including but not limited to: stable wind speed and direction (Varon et al., 2018), no outside sources intersecting the plume (Varon et al., 2018), no significant enhancement in the background concentrations (Irakulis-Loitxate et al., 2021), no pooling or other non-linear behavior within the plume (Bruno et al., 2024), or where to draw the boundaries of a plume (He et al., 2024). Similarly, large-scale chemical transport models tend to be quite stiff, with internal parameters and processes not being variable, even if on the subgrid scale this may introduce considerable uncertainty (Cohen and Prinn, 2011, Cohen and Wang 2014, Qin et al 2013; Hu et al., 2024; Lu et al., 2024).

D) Line 55: Airborne remote sensing is a highly technical and costly approach to record CH$_4$ fluxes from...

Where does the claim on "costly" come from? Actually, aerial approach has been widely employed due to its relative low cost comparing to other approaches. It will definitely less expensive to deploy than the method described in the current manuscript.

Airborne remote sensing requires the use of an aircraft, which is a very expensive item. Furthermore, many research teams do not have access to an aircraft. Additionally, such remote sensing likely requires an extensive number of disposables and people hours, adding to the cost. Since the reviewer assumes that our approach is more expensive, we are happy to share that we have spent far less for this entire experiment than it would cost us to just rent an aircraft for a single day.

In addition, aircraft based remote sensing makes many assumptions about the atmospheric column between the aircraft the surface source. How different thin clouds, aerosols, changes in surface reflectance, and other confounding factors are handled is highly technical. It also adds to the sources of model and parametric uncertainty.

We definitely feel that different sources all provide different benefits, and finding a way to use them all to the best possible extent is how we can move towards more solutions. Thank you for helping to support aircraft-based remote sensing. Please use this opportunity to the community also make better use of high frequency surface-based measurements.

E) Lines 61, 62: … but only after being calibrated by upward looking remotely sensed measurements…
Aren't almost all instruments need to be calibrated before adoption?

Yes, all instruments need to be calibrated before use. However, different calibrations are required under different atmospheric and geographic conditions. For example, the EM27 /SUN requires two instruments to be placed in the same position to observe data for one day to calibrate the instruments before each field measurement. However, the portable greenhouse gas analyzer used in this study has high accuracy and can only be calibrated by an engineer when the instrument equipment is purchased. It can be directly used for subsequent field experiments without the need for calibration, unless there are problems with the instrument during measurements and it needs to be sent back to the manufacturer for calibration again. However, in general, this instrument is relatively stable and less likely to have problems.

Lines 73-74: This work employs a high-frequency surface-based observation platform of CH$_4$ concentration, which is portable, economical, and unaffected by most

environmental factors

What is the proof for "economical"? Do not make any claims that you cannot support.

The economy of this method is relatively lower compared to observation methods such as airborne remote sensing and vehicle mounted navigation et al., based on our local conditions. For us, this approach is economically the most viable both because aircraft are highly limited or not possible to fly (as mentioned above), the instruments are relatively inexpensive to purchase because they are manufactured locally, and the number of people hours required to use and transport portable greenhouse gas analyzers is low. Perhaps in different places in the world this is seen differently?

F) Lines 78-79: Continuous 79 observations were made around known coal mines, unknown sources, and of background conditions.
Delete "of"?

Thank you for your suggestion. It has been revised in the manuscript.

Lines 80-90: High-frequency emissions calculated using these data were used to drive a 2-box model to attribute 81 emissions to the known mine and a second low production mine previously thought insignificant.
The model used the data to derive … not the data is used to derive the model.

Thank you for your suggestion. It has been revised in the manuscript.

G)    Lines 86-89. While the authors are talking about Changzi basin, why do we claim "province-wide" background are high?

Although the background concentration of methane in Shaanxi Province is indeed relatively high due to the large number of high gas coal mines and coal use industries, I agree with your suggestion. We have changed "province-wide" to "city-wide" as follows:

"Changzhi, Shanxi is located in a basin, with coal mines and associated coal use industries densely distributed throughout both flat central regions and around the mountainous edges (Figure 1), many of which are classified as high $CH_4$ emitting mines. Due to this combination, city-wide background $CH_4$ concentrations are very high and have large variation in time."

J)    Line 92: Observations were positioned along concentric…

Should be "instruments" be positioned, not observations

Thank you for your suggestion. It has been revised in the manuscript.

There are so many similar issues in the manuscript, and the reviewer will stop here at Page 5. The reviewer recommends having somebody proofread the English in the manuscript.

**References:**

Li, X., Cohen, J. B., Qin, K., Geng, H., Wu, X., Wu, L., Yang, C., Zhang, R., and Zhang, L., 2023, Remotely sensed and surface measurement-derived mass-conserving inversion of daily NOx emissions and inferred combustion technologies in energy-rich northern China: Atmospheric Chemistry and Physics, v. 23, no. 14, p. 8001-8019.

Qin, K., Hu, W., He, Q., Lu, F., and Cohen, J. B., 2024, Individual coal mine methane emissions constrained by eddy covariance measurements: low bias and missing sources: Atmospheric Chemistry and Physics, v. 24, no. 5, p. 3009-3028.

Qin, K., Lu, L., Liu, J., He, Q., Shi, J., Deng, W., Wang, S., and Cohen, J. B., 2023, Model-free daily inversion of NOx emissions using TROPOMI (MCMFE-NOx) and its uncertainty: Declining regulated emissions and growth of new sources: Remote Sensing of Environment, v. 295, p. 113720.

Wofsy, S. C., Goulden, M. L., Munger, J. W., Fan, S.-M., Bakwin, P. S., Daube, B. C., Bassow, S. L., and Bazzaz, F. A., 1993, Net Exchange of $CO_2$ in a Mid-Latitude Forest: Science, v. 260, no. 5112, p. 1314-1317.

**Reviewer #2:**

This paper discusses a method of estimating methane emissions from coal mines in China using in situ measurements at sites in all directions surrounding the facility at various distances. The authors also use a model for source attribution to determine which of the coal mines is impacting the measurement sites in cases where this is ambiguous.

The reviewer found this paper difficult to follow and much of the methodology was not as well described as it could be. Aspects of the measurements, uncertainty analyses and modelling are not well detailed which makes it difficult to assess the validity. The authors show a strong understanding of satellite work in the field of emissions and thus compare their results to this but there are very few references to other measurement work done, of which there is a significant amount. Based on my understanding of what was done I would not expect this to be as accurate a method of assessing emissions as other measurement based studies involving more continuous measurements or using aircraft or mobile measurement methods. Some of the assertions in the results of the paper are not sufficiently supported in this work though they may be accurate. There are several grammatical and spelling errors throughout the paper that should be corrected. I would recommend that this paper requires major revisions before being accepted to ACP.

**Major comments:**

Section 2.2: This section needs more detail on the operation of the instrumentation. How were the systems calibrated?

The portable greenhouse gas analyzer used in this study was calibrated by the company before first deployed to the field. During field experiments, each day in a background location we re-confirmed that the standard baseline was found within its reliable range. Overall, the instrument behaved stably throughout the experiment, and no significant problems were identified.

We appreciate the idea of using a standard zero air for calibration. However, we did not use this due to the fact that we could not find a commercially available sample air that replicates the high background $CH_4$ concentration levels and co-contaminants observed in Shanxi. Acknowledging this limitation, future studies will include additional observations using standardized gases, even if these gases do not fully represent the environmental conditions of the area being studied.

How did you ensure measurements were consistent when the location of an instrument was changed?

To ensure measurement consistency when the instrument's location was changed, after relocation, we rechecked the condition against the standard baseline. Furthermore, the instrument was allowed to stabilize and any air from the previous location was flushed out, with the first 10 minutes of measurement data discarded, to ensure that only local air was observed.

We have revised in the manuscript as follows:

"Atmospheric $CH_4$ concentrations at 5m above the surface were observed daily at 1 Hz from 8:30 to 17:00 local time in August 2022 using two portable greenhouse gas analyzers (LGR-915-0011, California, USA). Three different locations (1 km, 3 km and 5 km) were selected daily along a single direction from the CM-A, allowing a more consistent and precise calculation of the spatial gradient (Table 1). In order to reduce the time errors, using two portable greenhouse gas analyzers randomly selecting the three observation points during the daily measurements and without fixed sequence. During field experiments, each day in a background location we confirmed that the standard baseline was found within its reliable range. Further, to ensure measurement consistency when the instrument's location was changed, after relocation, we rechecked the condition against the standard baseline. Another, the instrument was allowed to stabilize and any air from the previous location was flushed out, with the first 10 minutes of measurement data discarded, to ensure that only local air was observed."

The sampling periods are very short, how did you ensure these are representative?

Despite the short measurement period, we adopted a careful measurement strategy when designing the experiment to ensure the maximum representativeness of the results given the limited time available for sampling. Specifically, we selected different directions and distances around the known coal mine to cover key areas of potential methane concentration changes. We carefully considered known sources from villages, coal chemical factories, coal washing areas, major highways as a source of potential leaking CNG vehicles, etc. By comprehensively measuring these areas, we were able to capture the main features of the spatial distribution. As pointed out in the paper, at one further site, we identified a very high value and re-confirmed a passing LNG truck at the time, providing robustness that our outliers make physical sense.

The reason we designed a two-week sampling period was it was the minimum time to capture the typical coal mine operation cycle and associated main activity patterns related to methane emissions (such as mining, high frequency meteorological variability, production peaks, weekend-weekday offsets, etc.). Therefore, although the data time is relatively short, it can still reflect the thinking from the perspective of what is considered the regularity of coal mining operations.

We acknowledge that the short measurement time may limit the long-term representativeness of the data, and in the future endeavor for longer case studies. However, short-term measurements are a common preliminary research method (Gorchov Negron et al. 2020, Shi et al. 2022), especially when resources and time are limited. This study provides important preliminary insights into the characteristics of methane emissions in coal mining areas. One such major finding is that there was an unidentified coal mine of significant emissions amount within the range of our study, which was not previously known and therefore would influence how a future longer-term study would be designed from the start. We also identified and statistically attributed high frequency emissions characteristics. In future studies, we plan to extend the measurement time and combine data from different seasons to further improve the broad applicability of the results.

Gorchov Negron, A. M., Kort, E. A., Conley, S. A., and Smith, M. L.: Airborne Assessment of Methane Emissions from Offshore Platforms in the U.S. Gulf of Mexico, Environ. Sci.Technol., 54, 5112-5120, https://doi.org/10.1021/acs.est.0c00179, 2020.

Shi, T., Han, Z., Han, G., Ma, X., Chen, H., Andersen, T., Mao, H., Chen, C., Zhang, H., and Gong, W.: Retrieving CH4-emission rates from coal mine ventilation shafts using UAV-based AirCore observations and the genetic algorithm–interior point penalty function (GA-IPPF) model, Atmos. Chem. Phys., 2, 13881-13896, https://acp.copernicus.org/articles/22/13881/2022/, 2022.

L55: I disagree with the statement that uncertainties are rarely assessed. There are many publications in the literature where uncertainties are thoroughly assessed for top-down emissions estimates. The two papers cited to support the claim that uncertainties are not assessed are over 10 years old.

Thank you for your words of caution. We acknowledge that the statement regarding uncertainty assessments may have been overly broad and insufficiently nuanced. Uncertainties consist of measurement uncertainties, model uncertainties, and parametric uncertainties. While it is true that recent years have seen a significant increase in studies that assess uncertainties in top-down emissions estimates, this work approaches consideration of these uncertainties on the model and parametric uncertainties themselves to some extent. First, this paper uses a more flexible and robust mass conserving approach, which allows the consideration of the change in wind speed, wind speed itself, the change in concentration, and concentration itself in tandem, without using background subtraction. Typical plume-based papers (Irakulis-Loitxate et al., 2021) do not consider how uncertainties in these variables may changes the size or boundaries of a plume (He et al., 2024), that pooling or concentration enhancement may occur within a plume, uncertainty in the background concentration, or that a plume may not be observable due to observational uncertainties operating along the spatial gradient across the edge of the plume. Furthermore, this work is attempting to introduce a flexible approach that is physically realistic but not as overly constrained as chemical transport models, and is still applicable to sub-grid scale variability. In doing this, our approach offers the consideration of a fuller range of uncertainties, with a particular emphasis on areas which have heavily polluted and mountainous conditions.

To address your concern, we have removed the less detailed statement and changed it to the following:

"Uncertainties in whether or not to consider the basic formulation of plume models as being reasonable under the observed conditions are rarely addressed holistically or in detail, including but not limited to: stable wind speed and direction (Varon et al., 2018), no outside sources intersecting the plume (Varon et al., 2018), no significant enhancement in the background concentrations (Irakulis-Loitxate et al., 2021), no pooling or other non-linear behavior within the plume (Bruno et al., 2024), or where to draw the boundaries of a plume (He et al., 2024). Similarly, large-scale chemical transport models tend to be quite stiff, with internal parameters and processes not being variable, even if on the subgrid scale this may introduce considerable uncertainty (Cohen and Prinn, 2011, Cohen and Wang 2014, Qin et al 2013; Hu et al., 2024; Lu et al., 2024)."

Irakulis-Loitxate, I., Guanter, L., Liu, Y.-N., Varon, D. J., Maasakkers, J. D., Zhang, Y., Chulakadabba, A., Wofsy, S. C., Thorpe, A. K., Duren, R. M., Frankenberg, C., Lyon, D. R., Hmiel, B., Cusworth, D. H., Zhang, Y., Segl, K., Gorroño, J., Sánchez-García, E., Sulprizio, M. P., Cao, K., Zhu, H., Liang, J., Li, X., Aben, I., and Jacob, D. J.: Satellite-based survey of extreme methane emissions in the Permian basin, Sci. Adv., 7, eabf4507, https://www.science.org/doi/abs/10.1126/sciadv.abf4507, 2021.

He, T.-L., Boyd, R. J., Varon, D. J., and Turner, A. J.: Increased methane emissions from oil and gas following the Soviet Union's collapse, PNAS, 121, p. e2314600121, https://doi.org/10.1073/pnas.2314600121, 2024.

Varon, D. J., Jacob, D. J., McKeever, J., Jervis, D., Durak, B. O. A., Xia, Y., and Huang, Y.: Quantifying methane point sources from fine-scale satellite observations of atmospheric methane plumes, Atmos. Meas. Tech., 11, 5673-5686, https://amt.copernicus.org/articles/11/5673/2018/, 2018.

Bruno, J. H., Jervis, D., Varon, D. J., and Jacob, D. J.: U-Plume: automated algorithm for plume detection and source quantification by satellite point-source imagers, Atmos. Meas. Tech., 17, 2625-2636, https://amt.copernicus.org/articles/17/2625/2024/, 2024.

Lu, L., Cohen, J. B., Qin, K., Li, X., and He, Q.: Identifying Missing Sources and Reducing NOx Emissions Uncertainty over China using Daily Satellite Data and a Mass-Conserving Method, EGUsphere [ACCEPTED], https://doi.org/10.5194/egusphere-2024-1903, 2024.

L123-126: This assumption is not commonly made in other papers using in situ measurements to do mass balance emissions estimates. Typically, measurements are taken upwind or out of a plume to determine a background concentration. All the citations in this section are for satellite-based emissions assessments, which this paper is not, so they are not the most appropriate comparison for this work.

Thank you for your thoughtful feedback. We agree that in situ studies typically determine background concentrations by measuring upwind or out of the plume. Actually, we also measured the background methane concentration in the area through this method.

What we aim to convey in this paragraph is that the practice of using the global methane background concentration as a reference to separate plumes may not be applicable to the study area in this paper. This is because the methane background concentration in our study area, as determined through field monitoring, is significantly higher than the global methane background concentration. Furthermore, there is substantial variability in the lowest site observed, as shown in Figure S2. We highlight the importance of accurately determining the range of concentrations in mass-balance emissions estimates, since our cite (as well as likely many others) do not conform to the simple assumption underlying your comment of a enhanced region in one direction and a stable background in another direction. The cited references all utilize the global latitudinal average methane background concentration for plume separation.

According to your comment, we have revised the manuscript as follows:

"These results highlight the importance of accurately determining background concentrations in mass-balance emissions estimates. Unlike satellite-based emissions assessments, which often rely on separating plumes from global latitude bands or climatological background states (Buchwitz et al., 2017; Irakulis-Loitxate et al., 2021; Lauvaux et al., 2022; Sadavarte et al., 2021), in situ measurements typically determine background concentrations by sampling upwind or outside the plume (Brantley et al. 2014). In this study, we have taken the spirit of the latter approach a step further, to

ensure that concentrations observed as relatively clean are actually representative of locally background air. This way ensures that our background concentrations were representative of the local conditions not influenced directly by the site of interest (Figure S2). This method provides a more accurate approach when the baseline itself also changes, as in the specific locations sampled in this work."

Buchwitz, M., Schneising, O., Reuter, M., Heymann, J., Krautwurst, S., Bovensmann, H., Burrows, J. P., Boesch, H., Parker, R. J., Somkuti, P., Detmers, R. G., Hasekamp, O. P., Aben, I., Butz, A., Frankenberg, C., and Turner, A. J.: Satellite-derived methane hotspot emission estimates using a fast data-driven method, Atmos. Chem. Phys., 17, 5751-5774, https://acp.copernicus.org/articles/17/5751/2017/, 2017.

Irakulis-Loitxate, I., Guanter, L., Liu, Y.-N., Varon, D. J., Maasakkers, J. D., Zhang, Y., Chulakadabba, A., Wofsy, S. C., Thorpe, A. K., Duren, R. M., Frankenberg, C., Lyon, D. R., Hmiel, B., Cusworth, D. H., Zhang, Y., Segl, K., Gorroño, J., Sánchez-García, E., Sulprizio, M. P., Cao, K., Zhu, H., Liang, J., Li, X., Aben, I., and Jacob, D. J.: Satellite-based survey of extreme methane emissions in the Permian basin, Sci. Adv., 7, eabf4507, https://www.science.org/doi/abs/10.1126/sciadv.abf4507, 2021.

Lauvaux, T., Giron, C., Mazzolini, M., d'Aspremont, A., Duren, R., Cusworth, D., Shindell, D., and Ciais, P.: Global assessment of oil and gas methane ultra-emitters, Science, 375, 557-561, https://www.science.org/doi/abs/10.1126/science.abj4351, 2022.

Sadavarte, P., Pandey, S., Maasakkers, J. D., Lorente, A., Borsdorff, T., Denier van der Gon, H., Houweling, S., and Aben, I.: Methane Emissions from Superemitting Coal Mines in Australia Quantified Using TROPOMI Satellite Observations, Environ. Sci. Technol., 55, 16573-16580, https://doi.org/10.1021/acs.est.1c03976, 2021.

Brantley, H. L., Thoma, E. D., Squier, W. C., Guven, B. B., and Lyon, D.: Assessment of methane emissions from oil and gas production pads using mobile measurements, Environ. Sci. Technol., 48(24), 14508-14515, https://pubs.acs.org/doi/full/10.1021/es503070q, 2014.

Section 2.4: The citations for the mass conserving approach proposed here all direct to previous satellite analyses but more information could be provided in this paper about how the approach is also applicable to in situ emissions that are not column based.

We have added in references with respect to reviewer 1. Actually, the equations we have used are the original equations, with assumptions, dating back to Cohen and Prinn, 2011, etc papers and even further back. They are the same equations used to solve for the forward and inverse versions of commonly used chemical transport models, including but not limited to WRF-CHEM and GEOS-CHEM. The major differences are first that we have swapped the spatial dimension for time, using the wind speed variable, and second that we have not included all of the second and third order physical driving terms. The equations reduce to the simple two-dimensional plume model assumption (that you are referencing) when the following conditions are all met: Emissions are steady in space and time, wind is steady in space and time, wind is non-divergent/non-convergent, there are no additional sources occurring within the plume, the background is always lower than the plume itself, and the background is not changing. We have also not accessed a three-dimensional version, and in the future it could make a challenging but interesting follow-up work.

A major difference occurs when the user wants to convert from our method's emissions given as ppm/time into a new unit's emissions of kg/time. Under this case, many assumptions are required about the size of the plume and the height of the plume, as observed by other species co-emitted with the methane plume species such as aerosols and water vapor show extreme variability, as demonstrated in Figure Res-1. Furthermore, the boundary layer itself is very complex over our region of interest due to the topographic variability (Guo et al., 2024). For this reason, the majority of our paper does not work using this variable, and we stick to ppm/min.

We believe that the computed emissions in this work demonstrate consistency by being re-run through the 2-box model and showing probabilistic constancy with the distribution of the observations. We believe that this new approach would add further support for other studies, and hope with further community improvement and application, to see it adapted more widely in the future.

Guo, J., Zhang, J., Shao, J., Chen, T., Bai, K., Sun, Y., Li, N., Wu, J., Li, R., Li, J., Guo, Q., Cohen, J. B., Zhai, P., Xu, X., and Hu, F.: A merged continental planetary boundary layer height dataset based on high-resolution radiosonde measurements, ERA5 reanalysis, and GLDAS: Earth Syst. Sci. Data, v. 16, no. 1, p. 1-14. https://essd.copernicus.org/articles/16/1/2024/essd-16-1-2024.html, 2024.

[Figure]

Figure Res 1. Photograph of typical coal mine plume

Section 2.5: Further details of the uncertainty analysis should be added. The section only states that uncertainty analysis was done and that the results assigned less than 5% to the input variables but does not describe how this was determined.

Thank you for pointing out the need to provide more details about the uncertainty analysis. We have taken the time to carefully go through the entirety of the datasets obtained, with the goal of analyzing the uncertainty in the observations themselves.

The first point is mathematical: the signal contains a real signal plus some amount of white noise, due to the observational uncertainty. Most papers refer to the uncertainty of the observations individually is at most 1% for the $CH_4$ observations (the uncertainty of portable greenhouse gas analyzers LGR-915-0011 < 1%) and 0.3% for the wind observations (Shi et al., 2022), in net far smaller than 5%. However, it is quite possible that the uncertainty in the region studied due to calibration issues may be larger. We wanted to be very conservative when we chose this number. Therefore, we went to the data itself. First, we acknowledge that any change of 30% or more in the sum of the change in the wind * derivative of concentration + concentration * derivative is considered to be emissions data, and therefore any change which is this large is already considered in the analysis with respect to the emissions itself. The second point is that there is uncertainty in the equation itself, and addressing when the actual equation itself when perturbated by the observations plus uncertainty may yield a value demonstrating actual emissions, as compared to noise which is being mis-represented as emissions. We have searched carefully (Tan et al., 2022)and cannot find a similar analysis applied to their representation of the change in height, wind speed derivative, and concentration derivative at minute-by-minute frequency.

As shown in plot Res-2 below, all of the data which is in the white noise region (i.e., the randomly occurring errors that occur throughout the dataset, when and where there is no emissions signal) are found to be approximately 5% and lower. It is for this reason that we have selected this value.

We believe that our approach to uncertainty analysis is reasonable and consistent. If the reviewers also want to see us use a larger uncertainty level, we could use a value of 10%, which is higher than all of the uncertainties except for those occurring during extremely high CH4 periods of time. However, based on our results accepted for publication in this other ACP article (Lu et al., 2025) we believe that the approach will not yield significantly different results.

[Figure]

Figure Res 2. An analysis of 3500 individual 1 minute frequency observations of $CH_4$ used in this work. The blue circles demonstrate the normalized data (data/maximum), while the red dots demonstrate the standard deviation divided by the mean of the normalized data. All data points which are counted as emissions are first filtered. The remaining red dots are considered the uncertainty, which in this case has most of the data with a value approximately equal to or smaller than 5%.

Shi, T., Han, Z., Han, G., Ma, X., Chen, H., Andersen, T., Mao, H., Chen, C., Zhang, H., and Gong, W.: Retrieving CH4-emission rates from coal mine ventilation shafts using UAV-based AirCore observations and the genetic algorithm–interior point penalty function (GA-IPPF) model, Atmos. Chem. Phys., 2, 13881-13896, https://acp.copernicus.org/articles/22/13881/2022/, 2022.

Tan, H., Zhang, L., Lu, X., Zhao, Y., Yao, B., Parker, R. J., and Boesch, H.: An integrated analysis of contemporary methane emissions and concentration trends over China using in situ and satellite observations and model simulations, Atmos. Chem. Phys., v. 22, no. 2, p. 1229-1249. https://acp.copernicus.org/articles/22/1229/2022/, 2022.

Lu, L., Cohen, J. B., Qin, K., Tiwari, P., Hu, W., Gao, H., & Zheng, B. Observational Uncertainty Causes Over Half of Top-Down NOx Emissions Over Northern China to Be Either Biased or Unreliable. (in review) Pre-print DOI: https://dx.doi.org/10.2139/ssrn.4984749

and that the results assigned less than 5% to the input variables but does not describe how this was determined.

We have discussed this above and have included a figure. One set of reasons that even the surface observations have a significant uncertainty is due to the fact that (a) the local concentration is far above standard calibration ranges, and that (b) this area also has a significant number of absorbing aerosols (Tiwari et al., 2025) and have a significant impact on the SWIR wavelengths used to detect $CH_4$.

Tiwari, P., Cohen, B. J., Lu, L., Wang, S., Li, X., Guan, L., Liu, Z., Li, Z., Qin, K.: A Synergistic, Multi-Platform Approach to Deriving Optically Constrained Aerosol Column Products: Insights into Spatio-temporal dynamics of Black Carbon around Xuzhou and Dhaka, 15 August 2024, PREPRINT (Version 1) available at Research Square [https://doi.org/10.21203/rs.3.rs-4859121/v1]

L400: Which emissions are considered in this average? All north and west sites, some subset? How can we equate the emissions calculated from the measurements at sites 1km from the site with those made at a site 5km away?

As explained above and in response to reviewer 1, the two box model equations account for the spatial gradients, which in turn are a function of the distance. Therefore, this is accounted for by the overall set of equations used herein. In theory, we could expand the two boxes to n-boxes, equally spaced and covering each observational site. Then the issue would resolve itself. This is what CTMs such as WRF or GEOS-CHEM do.

Though both may be downwind, the site further away will experience more dilution and thus will always have a lower emission rate calculated from this point unless you measure along a track downwind of the site rather than at a single point to ensure you are capturing all emissions. From the descriptions provided in the paper I do not understand how the rationale for combining emissions calculated from measurements at all the sites into one average.

Thank you for your thoughtful comments and questions. Below, I address your concerns in detail with respect to how we approximated this issue in this work.

The $CH_4$ emissions of Coal mine A reported in this study use all of the data from the north 1km $CH_4$ station, the $CH_4$ emissions of Coal mine B considered all of the data from the west 5km $CH_4$ station, and the background data came from all of the non-emissions data. We have revised the manuscript to be clearer, as follows:

*"The $CH_4$ emissions of Coal mine A reported in this study use all of the data from the north 1km $CH_4$ station, the $CH_4$ emissions of Coal mine B considered all of the data from the west 5km $CH_4$ station, and the background data came from all of the non-emissions data."*

In our study, the observation points used for the final $CH_4$ emission calculations were both located approximately 1 km from the respective coal mines (north 1km for Coal

Mine A and west 5km for Coal Mine B) (Figure 2, and copied below for clarity). Additionally, measurements were carefully selected based on wind direction to ensure $CH_4$ emissions were taken downwind of the plume. Since the distances are similar and all measurements were taken downwind of the plume, the issue of distance-based dilution does not occur. Therefore, we are confident that the calculated $CH_4$ emission from the coal mines without significant underestimation due to dilution.

[Figure]

**Figure 2. Locations of 2 individual coal mines (Green filled houses), a power plant (Red flag), and the 12 observation locations presented in this work (red double-outlined triangles). Distance from CM-A are given as concentric circles at 1km (blue), 3km (orange), and 5km (green).**

L418: I'm not sure it is fair to say this represents higher sampling diversity. The sampling frequency is higher but all measurements took place over just a 2-week period where satellites have a wider variety of measurements seasonally and annually.

Thank you for pointing this out. We agree that satellite observations, such as TROPOMI, provide broader spatial coverage, and due to its lifetime over the area of study, has a longer time series. However, over this area, the actual time series of available TROPOMI observations is shockingly smaller than expected than in other places of the world, as explained due to the challenging surface reflectance conditions, high absorbing aerosols loadings, and other issues. Recent work has established longer time series over this area, but insufficient to analyze seasonal or annual trends (Hu et al., CITE).

Our intention was not to suggest that the two-week in situ measurement period represents a more temporally diverse dataset overall. Instead, we aimed to highlight the advantage of minute-to-minute sampling frequency and its ability to capture short-term variations and extreme values that might not be detectable in daily or averaged satellite observations. This is especially the case, since the TROPOMI observations as well as

in-situ flux tower observations both demonstrate that the emissions are fat-tail/lognormally distributed, and therefore it is critical to observe the amount of both high and low emissions events in order to do a fuller characterization. To address your comment, we have revised the text in the manuscript to better clarify this point and to acknowledge the limitations of our study period.

Furthermore, we agree that longer time series should yield more precise results, and better explain the differences between typical, atypical, high, low and background types of conditions. However, we believe that the results presented herein are still consistent and meaningful. We have identified and attributed a previously unknown source. We have introduced a new approach and methodology which works under highly polluted and variable conditions. We believe that adaption of this approach will allow the community to have a new approach which can help globally with respect to issues of methane emission calculation, monitoring verification, and reporting. We certainly look forward to more community improvement. We are planning to add additional observations in the future, and look forward to updating the community at that time.

We have revised in the manuscript as follows:

"In this study, observations were made within 1km of the coal mines on a minute-to-minute basis, while the TROPOMI observed the $xCH_4$ over a space scale (5.5×7 $km^2$) and on a day-to-day average basis, the higher temporal resolution of our in-situ measurements offers an advantage in capturing short-term variations and extreme values within the study period when compared with TROPOMI's results, due to both the fat-tail distribution and the strong temporal variation of the observations."

**Minor Comments:**
L35: Should be reworded to say "Emissions from fossil fuel are one of the largest sources of anthropogenic methane"

Thank you for your suggestion. It has been revised in the manuscript.

L37: Should be reworded to say "Coal mines contribute up to X% of China's $CH_4$ emissions"

Thank you for your suggestion. It has been revised in the manuscript.

Figure 3: Line plots are not ideal for wind direction, would recommend using something else

Thank you for your suggestion. In order to present the information in the figure more clearly, we divided Figure 3 into Figure S1 and Figure S2. Similarly, we divided the same Figure 9 into Figure 10 and Figure 11, as shown below, and It has been revised in the manuscript.

[Figure]

**Figure S1.** Time series of CH$_4$ concentrations (ppm), wind speed (m s$^{-1}$) and wind direction ($^{\circ}$) measured at 1km (blue line), 3km (red line) and 5km (yellow line) located east of CM-A on two different days.

[Figure]

**Figure S2.** Time series of CH$_4$ concentrations (ppm), wind speed (m s$^{-1}$) and wind direction ($^{\circ}$) measured at 1km (blue line), 3km (red line) and 5km (yellow line) located south of CM-A on two different days.

[Figure]

**Figure 10.** Time series of CH$_4$ concentrations (ppm), wind speed (m s$^{-1}$) and wind direction ($^o$) measured at 1km (blue line), 3km (red line) and 5km (yellow line) located north of CM-A on two different days.

[Figure]

**Figure 11. Time series of CH₄ concentrations (ppm), wind speed (m s⁻¹) and wind direction (º) measured at 1km (blue line), 3km (red line) and 5km (yellow line) located north of CM-A on two different days.**

L135: Recommend showing the meteorological stations that were used on the map in Figure 2

Thank you for your suggestion. It has been revised as follow figure and revised in the manuscript.

[Figure]

**Figure 2. Locations of four individual coal mines (Green filled houses), a power plant (Red flag), and the 12 observation locations presented in this work (red double-outlined triangles). Distance from CM-A are given as concentric circles at 1km (blue), 3km (orange), and 5km (green).**

L138-139: Were all wind directions used to calculate the statistics? Wind directions are often unreliable when wind speeds are very low.

When using this model to calculate methane emissions, wind speed data was used. Wind direction data is used for data screening when conducting methane emission attribution analysis in the 2-box model. A flowchart of the application of the methane emission and attribution analysis model has been added to Section 2.6 of the article.

During times when very low wind speed was observed are not found to have an impact on the emissions calculated, as discussed more in-depth below in response to a different but similar question raised. Please see below for more details.

[Figure]

**Figure 7. Overview of the MCM2 and 2-box mass conserving model used in this work.**

L142: Please provide more information on how temperature and pressure were measured

We appreciate the reviewer's suggestion to provide more information on how temperature and pressure were measured. It has been revised in the manuscript as follows:

*"The temperature and pressure data were measured by a handheld meteorological instrument (HWS1000, ZOGLAB, China) with an accuracy of ± 0.5 °C for temperature and ± 0.5 hPa for pressure, ensuring reliable data collection. The meteorological instrument was calibrated according to the manufacturer's guidelines prior to use. Measurements were taken at 5 s intervals to capture temporal variations in the atmospheric conditions. The temperature and pressure data were averaged minute-by-minute to match observed wind data, and subsequently used to convert CH4 emissions unit (ppm min$^{-1}$) into policy-relevant unit (kg h$^{-1}$)."*

L181: What is u? (lower case u has not been defined)

Sorry, that's a mistake, it should be upper case U means wind sped (m s$^{-1}$). It has been revised in the manuscript.

L184: Would recommend a more recent reference

Thank you for your suggestion. We have incorporated a new reference into the manuscript as follows.

Conrad, B. M., Tyner, D. R., and Johnson, M. R.: Robust probabilities of detection and quantification uncertainty for aerial methane detection: Examples for three airborne technologies. Remote Sens. Environ., 288, 113499, https://doi.org/10.1016/j.rse.2023.113499, 2023.

Figure 6: Wind speeds are quite low here, what is the uncertainty in these wind directions due to the low wind speed?

During the time period that the wind speed is less than 0.9m/s, there were only three sets of emissions which are quantifiable, due to the rest of the data not meeting the minimum 30% change condition imposed by the methodology. The emissions computed during these times are found within the central 20th to 80th percentile of the net PDF of emissions from this cite, and therefore do not add bias to the resulting emissions distribution. If during these times, the instrument had an observational error more than 30% in the wind direction, then it may have influenced the emissions computed. We have not considered that the wind direction error may have been more than 30% during these times. However, since the emissions computed do not change the distribution, we do not believe that it would change the final values.

If we assume that these values are the result of a measurement error, then we should remove these points and re-compute the attribution analysis. We have found that the resulting emissions are from within the central portion of the distribution, and therefore that there is no change on the final results. This demonstrates the uniqueness and robustness of the approach herein, since there is not an assumed linear relationship between the wind-speed and emissions, and therefore there is no bias on the end results of excluding points which may have a higher chance of observational error (i.e., due to very slow wind speeds).

L249-251: Sentence is confusing

Thank you for your feedback. I am sorry for the original sentence might be unclear. Here's a revised version for better clarity and have been revised in the manuscript:

"The wind direction predominantly blew from CM-A towards the observation point (wind direction is between 150° and 210°), for about 60% of the daily observation time. Only one day (August 15) observed at 1km north with a significant amount of wind from the west (wind direction is between 240° and 300°), accounting for approximately 92.8% of the observation time on that day."

L296: Figure 10 does not have letters labelling the panels

Thank you for your suggestion. We have updated Figure 10 (now Figure 12) and Figure 11(now Figure 13) in the manuscript as follows.

[Figure]

**Figure 12.** Probability density map for CH₄ concentrations and wind rose measured at 1 km north (a), 1 km west (d), 3 km north (b), 3 km west (e), 5 km north (c), and 5 km west (f) of CM-A and corresponding wind rose chart.

[Figure]

**Figure 13.** Probability density map for CH₄ concentrations and wind rose measured at 1 km east (a), 1 km south (d), 3 km east (b), 3 km south (e), 5 km east (c), and 5 km south (f) of CM-A and corresponding wind rose chart.

L402: Suggest showing a figure of the fat tail distribution

There is not sufficient data to produce a PDF of the per hour emissions unit. However, we do have sufficient data to produce a PDF of the per minute emission results, which demonstrate a clear fat-tail distribution, as given below, and now in the updated paper as (Figure S3).

[Figure]

**Figure S3. Both coal mine A and coal mine B display a fat tail distribution.**

L419-421: Not really supported that this is the only likely reason why you did not see the expected distribution for the 2$^{nd}$ mine.

Thank you for pointing out that the sampling time might not be the only likely reason why the maximum CH$_4$ emissions at CM-B were smaller than the maximum estimated from TROPOMI. We agree with your assessment and have revised the manuscript to explicitly acknowledge additional potential reasons for this discrepancy.

"For this reason, it is likely that the sampling time (two days) at CM-B was insufficient to fully capture the fat tail of the CH$_4$ emissions. The estimation of CH$_4$ emissions from coal mine B in this study carries a significant uncertainty, and it should only be used to indicate that there are likely other substantial CH$_4$ emission sources within the study area. More accurate CH$_4$ emission would require additional monitoring data over a longer time. Furthermore, coal mine B has a relatively small coal production compared to coal mine A. Given that coal mine B is only about 6 km away from coal mine A and shares the same geological environment, its CH$_4$ emissions should theoretically be lower than those from coal mine A. However, the results from TROPOMI inversion show the opposite. This discrepancy is most likely due to the low spatial resolution of TROPOMI, which likely causes the CH$_4$ emissions from coal mine B to include emissions from coal mine A. "

L422-424: Only 2 coal mines were measured in this study. That's not enough data to make this claim.

Thank you for your suggestion. We acknowledge that only two coal mines were measured in this study, which is a limitation and does not provide a comprehensive dataset to establish a definitive relationship between production and $CH_4$ emissions. However, our intent was not to generalize this finding but to highlight that the observed emissions from these two coal mines align with the concept that higher production coal mines in geologically similar environments tend to emit more $CH_4$.

To clarify the misunderstanding, I have deleted this passage in the manuscript

L508: Recommend citing the chapter so people can easily find this information as it is the basis for the paper.

Thank you for your suggestion. We have cited the chapter and revised the manuscript accordingly, as shown below.

Brasseur, G. P., and Jacob, D. J.: Model Equations and Numerical Approaches, in: Modeling of atmospheric chemistry, Cambridge University Press Publishing, 84-91, https://books.google.com.hk/books, 2017.